# Self-Consuming Generative Models with Curated Data Provably Optimize Human Preferences

**Damien Ferbach**[1,2][*] **Quentin Bertrand**[1], **Avishek Joey Bose**[1,3], **Gauthier Gidel**[1,4]
[1]Mila, Université de Montréal [2]Ecole Normale Supérieure de Paris
[3]University of Oxford, [4] Canada CIFAR AI Chair

## Abstract

The rapid progress in generative models has resulted in impressive leaps in generation quality, blurring the lines between synthetic and real data. Web-scale datasets are now prone to the inevitable contamination by synthetic data, directly impacting the training of future generated models. Already, some theoretical results on self-consuming generative models (a.k.a., iterative retraining) have emerged in the literature, showcasing that either model collapse or stability could be possible depending on the fraction of generated data used at each retraining step. However, in practice, synthetic data is often subject to human feedback and curated by users before being used and uploaded online. For instance, many interfaces of popular text-to-image generative models, such as Stable Diffusion or Midjourney, produce several variations of an image for a given query which can eventually be curated by the users. In this paper, we theoretically study the impact of data curation on iterated retraining of generative models and show that it can be seen as an *implicit preference optimization mechanism*. However, unlike standard preference optimization, the generative model does not have access to the reward function or negative samples needed for pairwise comparisons. Moreover, our study doesn't require access to the density function, only to samples. We prove that, if the data is curated according to a reward model, then the expected reward of the iterative retraining procedure is maximized. We further provide theoretical results on the stability of the retraining loop when using a positive fraction of real data at each step. Finally, we conduct illustrative experiments on both synthetic datasets and on CIFAR10 showing that such a procedure amplifies biases of the reward model.

## 1 Introduction

Today state-of-the-art generative models can produce multi-modal generations virtually indistinguishable from human-created content, like text (Achiam et al., 2023), images (Stability AI, 2023), audio (Borsos et al., 2023), or videos (Villegas et al., 2022; Brooks et al., 2024). The democratization of these powerful models by open-sourcing their weights (Stability AI, 2023; Jiang et al., 2023; Touvron et al., 2023) or allowing public inference (Ramesh et al., 2021; Midjourney, 2023; Achiam et al., 2023) paves the way to creating synthetic content at an unprecedented scale—the results inevitably populate the Web. In particular, existing datasets are already composed of synthetic data such as JourneyDB (Pan et al., 2023) and SAC (Pressman et al., 2022). Moreover, Alemohammad et al. (2024, Fig. 2) showed LAION-5B (Schuhmann et al., 2022), a large-scale Web-crawled dataset used to train future generative models, *already* contains synthetically generated images.

There is strong evidence that the synthetic data on the web has been, to a large extent, curated by human users. For instance, the LAION-Aesthetics datasets contains synthetically generated

---

[*]Corresponding author: `ferbach.damien@gmail.com`

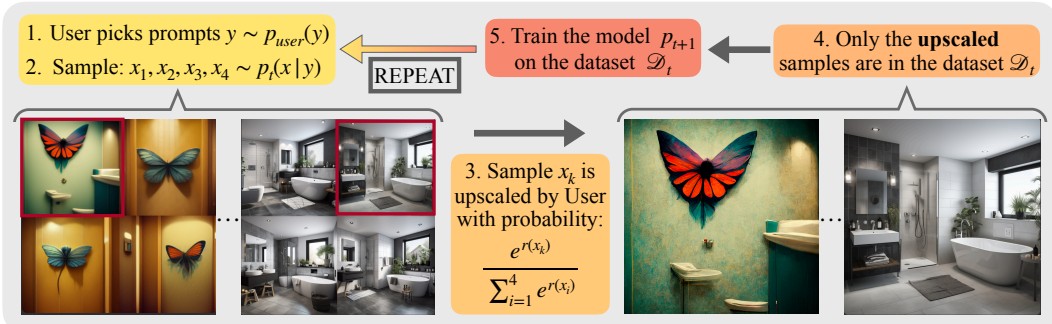

Figure 1: Illustration of the curation phenomenon: 1. User proposes prompts such as "butterfly going to the bathroom", 2. Four images are generated with Midjourney, 3. User only *upscale* one (e.g. the top left image) image, 4. Solely upscaled images are incorporated into the JourneyDB dataset (Pan et al., 2023). Samples from other diffusion models can be found in Figures 12a and 12b.

images that have been curated using a reward model learned from human feedback on the Simulacra Aesthetic Captions dataset (Pressman et al., 2022). Additionally, the JourneyDB dataset, contains human-picked images from the Midjourney (Midjourney 2023) discord server, that have been *upscaled*, *i.e.,* images that have been requested in a higher resolution (see Figure 1). More generally, the user interface of the most popular state-of-the-art text-to-image models (*e.g.,* Midjourney and Stable Diffusion 2.1 Huggingface implementation) proposes four alternatives for a single prompt for the user to pick their favorite.

While the consequences of iterative retraining of generative models on synthetically generated data have raised a lot of attention in the community (Alemohammad et al., 2024; Shumailov et al., 2023; Bertrand et al., 2024; Dohmatob et al., 2024a), previous works do not consider that generated data could be curated. This subtle nuance may be of major importance. Indeed, in numerous applications, augmenting the datasets with curated synthetically generated data is found to enhance the performance of the downstream task (Azizi et al., 2023; Wang et al., 2023; Gillman et al., 2024) and even generative models themselves (Hemmat et al., 2023; Gulcehre et al., 2023), hinting that quality might not be the biggest problem. On the other hand, recent works Wyllie et al. (2024); Chen et al. (2024b) showed that this might lead to new issues, such as bias amplification.

This is why, in this work, we aim to theoretically understand how the process of curation of synthetic data is connected with the reward model underlying human preferences and what distribution is learned by generative models trained on such curated synthetic data.

**Main contributions**. We summarize our core contributions as the following:

- We first focus on the self-consuming loop on (only) curated synthetic samples: we show that the expected reward gets maximized in Lemma 2.2 and that its variance vanishes. We further provide a convergence result in Theorem 2.1.
- We additionally study the iterative retraining loop when real data is re-injected at each step: we first improve previous results of stability using raw synthetic samples by Bertrand et al. (2024) and show convergence in Kullback-Leibler (KL) divergence to the optimal distribution Theorem 2.2. When using curated synthetic samples, we show that the KL divergence with the optimal distribution remains bounded Theorem 2.4, as well as an improvement on the expected reward Theorem 2.3, enlightening connections with Reinforcement Learning from Human Feedback (RLHF).
- We finally illustrate our theoretical results on synthetic datasets (mixtures of Gaussians and two moons) as well as natural images on CIFAR10 in Section 4. We highlight how curation based on the confidence of a classifier can lead to the emergence of biases (Wyllie et al., 2024).

## 2   Iterative retraining with curated synthetic data

We now study the fully synthetic self-consuming loop with curated samples. Unlike previous works that do not take curation into account (Alemohammad et al., 2024; Shumailov et al., 2023) and focused on stability of the process (Bertrand et al., 2024), we show that retraining with curated samples both maximizes an underlying reward whose variance collapses, and converges to maximum reward regions. In Section 2.1 we first specify explicitly the distribution induced

by a discrete choice model and highlight connections with RLHF. We additionally show that the expected reward increases and that its variance vanishes Lemma 2.2. Finally, inspired by stability results of Bertrand et al. (2024), in Section 2.2 we extend our study to settings when real data is injected. More precisely, we improve previous results of stability of Bertrand et al. (2024) without curation and provide novel theoretical bounds when the synthetic data is curated.

**Notation and conventions**. Lowercase letters $p$ denote densities while uppercase letters $\mathbb{P}$ indicate their associated probabilities. We denote $p_{\text{data}} \in \mathcal{P}(\mathbb{R}^d)$ the real data distribution and for $t \in \mathbb{N}$, we denote $p_t \in \mathcal{P}(\mathbb{R}^d)$ the model distribution at step $t$ of the iterative retraining loop. Analogously, $\theta_t$ is the corresponding parameters of the learned parametric model $p_t$. We write $p_0$ to indicate the initial model learned using maximum likelihood on $p_{\text{data}}$, this includes many modern generative model families such as diffusion models (Ho et al., 2020; Song et al., 2021) and flow-matching methods (Lipman et al., 2022; Tong et al., 2023b).

**Discrete $K$-choice model**. We propose using a discrete choice model for the curation phenomenon illustrated in Figure 1, where users pick their preferred image that will be upscaled in the next dataset. Modeling human preferences is usually done via the Luce choice rule model (Shepard, 1957; Luce et al., 1963; Dumoulin et al., 2023), where the human is modeled as a rational Bayesian subject. The probability that $x_1$ is preferred to $x_2$ is formulated using an underlying reward function $r(x)$ and Plackett-Luce model (equivalently Bradley-Terry model) (Bradley and Terry 1952). Under Luce's choice axiom (Luce, 1959), it is possible to generalize this formula to $K \geq 1$ samples as in Equation 2. For given samples $x_1, \ldots, x_K$ drawn from $p_t$, the random curated sample denoted $\hat{x}$ is chosen according to this Plackett-Luce model $\hat{x} \sim \mathcal{PL}(x_1, \ldots, x_K)$ as in Equation 2 (Bradley and Terry, 1952; Luce, 1959; Plackett, 1975). In particular, the curation procedure can be summarized as follows

$$
\begin{aligned}
&\text{1) Sample } x_1 \sim p_t, \ldots, x_K \sim p_t \,, \text{ independently,} &&(1)\\
&\text{2) Pick } \hat{x} \sim \mathcal{PL}(x_1, \ldots, x_K) \,, \text{ i.e., } \mathbb{P}(\hat{x} = x_k | x_1, \ldots, x_K) = \frac{e^{r(x_k)}}{\sum_{j=1}^K e^{r(x_j)}} \,, 1 \leq k \leq K. &&(2)
\end{aligned}
$$

**Self-consuming loop**. After generating and curating a synthetic dataset according to Equations 1 and 2, the next generation of generative models is trained either solely on the distribution of curated samples ($\lambda \to \infty$), or on a mixture of reference samples (that either comes from real data $p_{\text{data}}$ or a reference generative model $p_0$) and synthetic curated samples ($\lambda < \infty$) depending on the studied setting

$$
p_{t+1} = \arg\max_{p \in \mathcal{P}} \frac{1}{1+\lambda} \cdot \mathbb{E}_{x \sim p_{\text{ref}}} \left[ \log p(x) \right] + \frac{\lambda}{1+\lambda} \cdot \mathbb{E}_{\substack{x_1, \ldots, x_K \sim p_t \\ \hat{x} \sim \mathcal{PL}(x_1, \ldots, x_K)}} \left[ \log p(\hat{x}) \right] \,. \tag{3}
$$

where $\mathcal{P}$ is the set of achievable distributions with our model. This work aims to study the retraining dynamics of the distribution defined in Equation 3. First, in Section 2.1 we study the simplified dynamics of Equation 3 in the regime $\lambda \to \infty$, *i.e.,* when solely retraining on curated synthetic data and show convergence of the process but variance collapse. In Section 2.2 we study the exact dynamics given in Equation 3 and the impact on the stability of retraining on a mix of real data synthetic curated data.

## 2.1 Iterative retraining only on the curated synthetic samples

In this section, we study the dynamics of the density learned through iterative discrete $K$-choice curation in the fully-synthetic setting (*i.e.,* $\lambda \to \infty$): Equation 3 boils down to

$$
p_{t+1} = \arg\max_{p \in \mathcal{P}} \mathbb{E}_{\substack{x_1, \ldots, x_K \sim p_t \\ \hat{x} \sim \mathcal{PL}(x_1, \ldots, x_K)}} \left[ \log p(\hat{x}) \right] \,. \tag{4}
$$

As a warm-up, we first consider the limit of $K \to \infty$ in Lemma 2.1 and draw explicit connections with RLHF. This simplification yields a closed-form formula form for the solution of Equation 4 and provides intuitions for the dynamics of learning on curated samples.

**Lemma 2.1.** *Let $p_{t+1}$ be defined as in Equation 4. If $\mathcal{P} = \mathcal{P}(\mathbb{R}^d)$ is the set of probability distributions on $\mathbb{R}^d$, and if we assume that $\mathbb{E}_{y \sim p_t}\left[e^{r(y)}\right] < \infty$, then we have for all $x \in \mathbb{R}^d$,*

$$p_{t+1}(x) \xrightarrow{K \to \infty} p_t(x) \frac{e^{r(x)}}{\mathbb{E}_{\tilde{x} \sim p_t}\left[e^{r(\tilde{x})}\right]} \quad . \tag{5}$$

**Dependency on $K$ and connection to RLHF.** The proof of Lemma 2.1 relies on the fact that we can obtain a closed-form formula for the density $p_{t+1}$ induced from discrete $K$-choice curation on $p_t$ (Equation 4). This is done in Appendix A.4.1 where we show that its density can be written

$$p_{t+1}(x) = p_t(x) \cdot H_{p_t}^K(x) \,, \text{ with } H_{p_t}^K(x) := \mathbb{E}_{x_1, \dots, x_{K-1} \sim p_t}\left[\frac{K \cdot e^{r(x)}}{e^{r(x)} + \sum_{i=1}^{K-1} e^{r(x_i)}}\right] \,. \tag{6}$$

The latter directly implies that for all $K \geq 1$, $H_{p_t}^K(x) \in (0, K)$. In particular, small values of $K$ act as a *regularization* which prevents the density from blowing up too much in high rewards areas. On the other hand, the higher the number of samples used for curation, the more it can affect the induced distribution. In the limit $K \to \infty$, Lemma 2.1 shows an interesting connection between iterative retraining on curated data and reward maximization via RLHF. Given a supervised-finetuned model distribution $\pi^{\text{SFT}}$ and a regularization parameter $\beta$, the goal of RLHF is to find a policy that maximizes a reward $r(x)$ fitted on human preferences :

$$\pi^{\text{RLHF}} = \arg\max_{\pi} \mathbb{E}_{x \sim \pi}\left[r(x)\right] - \beta D_{\text{KL}}\left(\pi || \pi^{\text{SFT}}\right) \,, \text{ which has a closed form formula},$$

$$\pi^{\text{RLHF}}(x) \propto \pi^{\text{SFT}}(x) e^{r(x)/\beta} \quad \text{(Go et al., 2023; Rafailov et al., 2024)}.$$

Therefore, in the limit $K \to \infty$, Equation 5 shows that performing iterative retraining with human curation for $t$ iterations is equivalent to performing RLHF with hyperparameter $\beta = \frac{1}{t}$ from the initial distribution $\pi^{\text{SFT}} := p_0$. The corresponding regularization parameter $\beta$ is inversely proportional to the number of retraining steps. This connection is surprising since performing maximum likelihood on a curated distribution (Equation 3) is a priori different than directly maximizing a reward with Kullback-Leibler (KL) regularization.

To prove that curation both increases the expected reward and reduces the variance, we will need an additional assumption that the reward is bounded at initialization. We decompose this assumption into three sub-assumptions of increasing thrength:

**Assumption 2.1.**       A. *The distribution $p \in \mathcal{P}(\mathbb{R}^d)$ has a density w.r.t. Lebesgue measure and $\mathbb{E}_p[e^{r(x)}] < \infty$.*

    B. *The distribution $p \in \mathcal{P}(\mathbb{R}^d)$ has a density w.r.t. Lebesgue measure and there exists $r_* \in \mathbb{R}$ such that: (a) $p$-almost surely, $r(x) \leq r_*$ and (b) $p$ puts positive mass in a neighborhood of $r_*$ i.e., $\forall \varepsilon > 0, \mathbb{P}(r(x) \geq r_* - \varepsilon) > 0$.*

    C. *Assum. 2.1.B and $\mathbb{P}(r(x) = r^*) > 0$.*

In particular, Assumption 2.1 A and B are satisfied if we suppose that the reward bounded, which is reasonable if we suppose it is continuous given that the set of images $[0, 1]^d$ is compact. Note that assuming (a), we can always choose $r_*$ such that (b) is satisfied by picking the smallest value that almost surely bounds the reward at initialization. On the other hand (b) imposes that $r_*$ is the smallest value that a.s. upper-bounds the reward. In a nutshell, $r_*$ should be thought as the smallest number that upper-bounds the random variable $p_0$ with probability 1. For example, a Uniform distribution on the interval $[0, 10]$, $r_* = 10$ whereas for unbounded distributions such as $\mathcal{N}(0, 1)$, $r_*$ does not exist. In other words, $r_* = \inf\{r \in \mathbb{R}, \mathbb{P}_0(r(x) \leq r_*) = 1\}$. This shows that $r_*$ is uniquely defined which is an important point as we will show convergence of $p_t$ towards the level set $r(x) = r_*$ in Lemma 2.2 and Theorem 2.1.

Lemma 2.2 states the reward expectation increases proportionally to the reward variance.

**Lemma 2.2.** *Let $p_{t+1}$ be the distribution induced from a discrete choice model on $p_t$ (Equation 4). Suppose Assumption 2.1 B holds, then the expected reward increases proportionally to its variance at each retraining iteration:*

$$\mathbb{E}_{p_{t+1}}\left[e^{r(x)}\right] \geq \mathbb{E}_{p_t}\left[e^{r(x)}\right] + \frac{K-1}{K}\frac{\text{Var}_{p_t}\left[e^{r(x)}\right]}{e^{r_*}} \quad . \tag{7}$$

*Especially the expected reward converges to the maximum reward and its variance vanishes:*

$$\mathbb{E}_{p_t}\left[e^{r(x)}\right] \xrightarrow{t\to\infty} e^{r_*} \quad and \quad \text{Var}_{p_t}\left[e^{r(x)}\right] \xrightarrow{t\to\infty} 0 \quad .$$

**Discussion**. Lemma 2.2 shows that the reward augmentation is directly proportional to the reward variance at each retraining step. In other words, the more heterogeneous the reward is, the more its expectation increases at the next step. Lemma 2.2 further shows that the expected reward converges towards the reward maximizers. We can additionally deduce that the variance is doomed to vanish. This is detailed in Appendix A.4.3 which additionally states that the reward variance decreases fast enough to have finite sum. Finally, we note that Lemma 2.2 helps us understand the fixed points of this process: due to the variance term in Equation 7, a fixed point of the retraining loop must put mass on a single level set of the reward function. The reciprocal is obviously true as detailed in the appendix (Lemma A.3).

We can finally show a stronger result of convergence for the Kullback-Leibler divergence. We will need to assume that at initialization, the probability density puts a positive mass on the level set $r(x) = r_*$. This corresponds to Assumption 2.1 C. Without this assumption, the probability density support would consecutively vanish towards the maximizer of the reward preventing KL convergence. Under assumption 2.1 C, we can denote $p_*$ the probability density at initialization restricted to the domain that maximizes the reward and renormalized: $p_*(x) := \frac{p_0(x)\mathbb{1}_{r(x)=r_*}}{\mathbb{P}_0(r(x)=r_*)}$.

**Theorem 2.1.** *Let for all $t \geq 0$, $p_{t+1}$ be the distribution induced from a discrete choice model on $p_t$ (Equation 4) where $\mathcal{P} = \mathcal{P}(\mathbb{R}^d)$ is the set of probability distributions on $\mathbb{R}^d$. If $p_0$ satisfies Assumption 2.1 C, then we can define $p_*(x) := \frac{p_0(x)\mathbb{1}_{r(x)=r_*}}{\mathbb{P}_0(r(x)=r_*)}$ and the self-consuming loop on curated samples $p_t$ converges to $p_*$:*

$$D_{\text{KL}}(p_*||p_t) \xrightarrow{t\to\infty} 0.$$

Theorem 2.1 proved in Appendix A.4.5 shows that the process of retraining with curation Equation 2 eventually converges to *the highest level set of the reward reached at initialization*. In particular, in the limit of a large number of retraining steps, the probability of all smaller rewards vanishes. This can have strong implications when retraining the next generation of generative models on a curated Web-scaled dataset: the learned distribution will lose diversity and collapse to the highest reward samples.

## 2.2 Stability of iterative retraining on a mixture of real and synthetic data

After showing convergence but variance collapse of the self-consuming loop on curated synthetic samples, we now study the impact on the stability of injecting real data at each step. This setting is motivated by the recent work of Bertrand et al. (2024) that showed stability of the iterative retraining loop with real and synthetic data around a local maximizer $\theta_*$ of the training distribution likelihood. This setting is furthermore relevant since Web-scrolled datasets will presumably keep containing a mixture of real data and human-curated synthetic data. In Section 2.2.1 we first improve previous results on retraining on mixed datasets which underlines the beneficial impact of real data on stability and in Section 2.2.2, we prove both stability and reward augmentation in the setting of mixed real and *curated* synthetic data.

### 2.2.1 Iterative retraining without curation

To motivate the impact of real data on the stability of the retraining loop with curation, we focus first on its impact without curation and improve previous results in that setting in Theorem 2.2.

**Setting**. In this section only, following the approach of Bertrand et al. (2024), we will not assume infinite capacity for our distribution (*i.e.*, $\mathcal{P} \neq \mathcal{P}(\mathbb{R}^d)$ and hence adopt a parametric approach $\mathcal{P} = \mathcal{P}_\Theta := \{p_\theta \mid \theta \in \Theta\}$. Given the current generative model distribution $p_{\theta_t}$, $p_{\theta_{t+1}}$ must at the next iteration maximize the combined log-likelihood of real and generated data with hyperparameter $\lambda$, *i.e.*, Equation 3 becomes:

$$p_{\theta_{t+1}} = \arg\max_{p_\theta \in \mathcal{P}_\Theta} \frac{1}{1+\lambda} \cdot \mathbb{E}_{p_{\mathrm{data}}}[\log p_\theta(x)] + \frac{\lambda}{1+\lambda} \cdot \mathbb{E}_{p_{\theta_t}}[\log p_\theta(x)] \ .$$

We finally denote $p_{\theta_*} = \arg\max_{p_\theta \in \mathcal{P}_\Theta} \mathbb{E}_{p_{\mathrm{data}}}[\log p_\theta(x)]$ a maximizer of the data distribution log-likelihood. We also make the following assumption taken from Bertrand et al. (2024):

> **Assumption 2.2.** *For $\theta$ close enough to $\theta_*$, the mapping $x \mapsto \nabla_\theta^2 \log p_\theta(x)$ is $L$-Lipschitz and the mapping $\theta \mapsto \mathbb{E}_{p_{\mathrm{data}}}[\log p_\theta(x)]$ is continuously twice differentiable with $\mathbb{E}_{p_{\mathrm{data}}}[\nabla_\theta^2 \log p_\theta(x)] \preceq -\alpha I \prec 0$. Further suppose $W_1(p_{\theta_*}, p_{\mathrm{data}}) \leq \varepsilon$, i.e. $p_{\theta_*}$ is close to the data distribution $p_{\mathrm{data}}$.*

Bertrand et al. (2024) proved stability of the retraining loop provided $\lambda$ is sufficiently small. However, their proof is restricted to $\lambda < \frac{1}{2}$, preventing the use of a fraction of synthetic data $\frac{\lambda}{1+\lambda}$ bigger than one-third which they left as future work. In Theorem 2.2, we extend their proof to any fraction of synthetic data provided the best model distribution is sufficiently close to $p_{\mathrm{data}}$ in Wasserstein distance (Villani et al., 2009) *i.e.*, $\mathcal{W}_1(p_{\theta_*}, p_{\mathrm{data}}) \leq \varepsilon < \frac{\alpha}{L}$. Additionally, we express the result in distribution, while they expressed it in parameter space.

> **Theorem 2.2.** *Under Assumption 2.2, if $L\varepsilon < \alpha$ and $\lambda < \frac{\alpha}{2L\varepsilon}$, then there exists a neighborhood of the optimal distribution parameters $\theta_*$ such that for any initial parameters $\theta_0$ in that neighborhood, $p_{\theta_t}$ converges to $p_{\theta_*}$ exponentially fast:*
> $$D_{\mathrm{KL}}(p_{\theta_*} || p_{\theta_t}) = \tilde{\mathcal{O}} \left( \left( \frac{\lambda(\alpha + \varepsilon L)}{\alpha + \lambda(\alpha - \varepsilon L)} \right)^{2t} \right) \ .$$

### 2.2.2 Iterative retraining on a mixture of real and curated samples

Interestingly when curating the synthetic samples we cannot expect stability around the optimal distribution ($\theta_*$ in Theorem 2.2) since it is no longer a fixed point of the retraining loop. We will instead show a closeness result in KL divergence combined with an increasing property of the expectation of the reward, which bears close connections to RLHF. We therefore now study the setting described in Equation 3 where the synthetic samples are curated using a discrete $K$-choice model and real data is reused at each step ($\lambda < \infty$). In other words, we suppose that the retraining step uses a mixture of a reference distribution and a curated distribution as

$$p_{t+1}(x) = \frac{1}{1+\lambda}p_{\mathrm{ref}}(x) + \frac{\lambda}{1+\lambda}p_t(x) \cdot H_{p_t}^K(x) \qquad (H_{p_t}^K \text{ is defined in Equation 6}) \ . \qquad (8)$$

In Theorem 2.3, we prove that when retraining on a mixture of curated samples and samples from the reference distribution, the reward increases with respect to the reference distribution:

**Theorem 2.3.** *Let* $\lambda > 0$ *and consider the process* $(p_t)$ *defined in eq. 8, with* $p_0 = p_{\text{ref}}$. *If* $p_{\text{ref}}$ *satisfies Assumption 2.1 B, then for all* $t \geq 1$:

$$\mathbb{E}_{p_t}\left[e^{r(x)}\right] \geq \mathbb{E}_{p_{\text{ref}}}\left[e^{r(x)}\right] + \frac{\lambda}{(1+\lambda)^3}\frac{(K-1)\text{Var}_{p_{\text{ref}}}\left[e^{r(x)}\right]}{Ke^{r_*}} \quad .$$

**Discussion**. A first interesting case is taking the reference distribution $p_{\text{ref}}$ equal to $p_{\text{data}}$. In that case, we recover the fact that $p_{\text{data}}$ is not a fixed point of the retraining loop as soon as different reward values have non-zero probabilities to happen (we recover the result from Lemma A.3). In fact, Theorem 2.3 shows that such a process initialized at $p_{\text{data}}$ will increase the reward expectation. The second interesting case is taking $p_{\text{ref}} = p_0$ the generative model at initialization. In that case, retraining on a mixture of samples from the initial model and curated samples from the current model improves the reward expectation with respect to initialization.

After showing that such a retraining loop improves the expected reward, we can conversely show that this process does not deviate too much from $p_{\text{ref}}$.

**Theorem 2.4.** *Let* $\lambda > 0$ *and* $p_{\text{ref}} \in \mathcal{P}(\mathbb{R}^d)$ *with a density with respect to Lebesgue measure. Consider the process* $(p_t)$ *defined in Equation 8, with* $p_0 = p_{\text{ref}}$. *Suppose that* $\lambda < \frac{1}{K-1}$, *then, for all* $t \geq 1$

$$D_{\text{KL}}(p_t||p_{\text{ref}}) \leq -\log\left(1 - \lambda(K-1)\right) \quad .$$

Applying Theorem 2.4 with $p_{\text{ref}} = p_{\text{data}}$ shows that retraining on a mixture of real and curated synthetic samples does not deviate too much from the data distribution. On the other hand, when setting $p_{\text{ref}}$ to be any initial model distribution, we see that reusing samples from the initial model stabilizes the retraining loop around initialization.

**Connection with RLHF**. Theorem 2.3 and Theorem 2.4 together emphasize that retraining on a mixture of reference and filtered synthetic data bears important connections with RLHF. Indeed, the RLHF objective is composed of both a reward maximization term and a KL regularization between the current and initial model. In turn, Theorem 2.3 states that the expected reward increases and Theorem 2.4 shows that the KL divergence with respect to initialization remains bounded. The upper bound on the KL divergence further indicates that setting $K$ small, *i.e.,* using fewer samples for comparison acts as a regularizer, as previously noticed.

# 3 Related work

**Iterative retraining on synthetic data and model collapse**. The study of the retraining loop of a generative model on synthetic data has witnessed a recent surge of interest. Alemohammad et al. (2024); Shumailov et al. (2023) first evidenced catastrophic degradation of the generated data in the fully synthetic loop. Bertrand et al. (2024) mitigate these conclusions in the setting where the model is retrained on a mixture of synthetic and real data and they show the stability of the process around the data distribution. Briesch et al. (2023) specifically focus on large langage models and Hataya et al. (2023); Martínez et al. (2023) study large scale datasets. A recent theoretical push by Dohmatob et al. (2024a,b) provides bounds on the performance degradation in the regression setting as well as modified scaling laws. Finally recent works, Wyllie et al. (2024); Chen et al. (2024b) study the emergence or amplification of biases in self-consuming loops.

**Aligning models with human preferences**. With the urgent and critical safety concerns of public deployment, the need to align models with human preferences has gained significant importance. RLHF is a popular reinforcement learning technique to align an already pretrained and finetuned model on human preferences (Christiano et al., 2017; Stiennon et al., 2020; Lee et al., 2021; Ouyang et al., 2022; Shin et al., 2023). It consists of two steps: first fitting a reward $r(x)$ on human preferences using a dataset of pairwise human comparisons and then, maximizing the expected reward over the model distribution. A Kullback-Leibler regularization to the initial model is further used during the maximization step to avoid reward hacking (Skalse et al., 2022; Chen et al., 2024a) or catastrophic forgetting (Korbak et al., 2022). Variants of RLHF have recently been proposed such as Direct Preference Optimization (DPO) which maximizes the reward directly without modeling

it (Rafailov et al., 2024), Identity Preference Optimization (IPO) (Azar et al., 2024) or Kahneman-Tversky Optimization (KTO) (Ethayarajh et al., 2024).

# 4 Experiments

This section aims to empirically illustrate our previous theoretical results on how curation impacts the self-consuming loop. In Algorithm 1, we recall and detail the different steps performed in our experiments.

**Synthetic datasets**. We first focus on two synthetic datasets: a mixture of Gaussians and the two moons dataset. For both datasets, we study the two settings of solely retraining on curated synthetic samples ($\lambda = \infty$) and mixed datasets ($\lambda = 1$). In Figure 4, we iteratively retrain a denoising diffusion probabilistic model (DDPM, Ho et al. 2020) on a mixture of $8$ Gaussians. The reward $r(x)$ used for the discrete choice model is the clipped negative Euclidean distance to one of the centers of the Gaussians $x_*$, *i.e.*, $r(x) := -\gamma \max\{0, \|x - x_*\| - r_{\min}\}$ where we choose $\gamma = 10, r_{\min} = 1$. Clipping the distance is used to ensure that the process does not collapse to a single point. Indeed applying Theorem 2.1, we know that the density will converge to a renormalized Gaussian distribution restricted to the ball centered at $x_*$ of radius $r_{\min}$. In Figure 5, we plot the retraining curves on the two moons dataset: to compute the reward, we use an MLP classifier with 2 hidden layers of width $512$ which yields probabilities $q_0(x), q_1(x)$ for each class. The reward is then defined as : $r(x) := \gamma q_0(x), \gamma > 0$. Both Figure 4 and Figure 5 illustrate that retraining on solely curated samples induces collapse to regions that maximize the reward: respectively one mode of the MoG or one single moon. On the other hand, the use of real data results at the same time both in stability and higher density in high reward regions. Further experimental details are provided in Appendix A.5.

## 4.1 Natural images on CIFAR10

**Setting**. We train a normalizing flow using optimal transport conditional flow matching (Lipman et al., 2022; Shaul et al., 2023; Tong et al., 2023b) with the *torchcfm* library Tong et al. (2023a, 2024). The initial model has been pretrained on the $50000$ train images of the CIFAR-10 dataset (Krizhevsky et al., 2009). At each iteration, we generate $5 \cdot 10^4$ samples using the current model from which we keep $2.5 \cdot 10^3$ samples filtered by discrete $K$-choice comparisons. The reward $r(x)$ is computed using the class probabilities $q_0(x), \ldots, q_9(x)$ from a *pretrained* VGG11 classifier (Simonyan and Zisserman, 2014) with $92.39\%$ test accuracy. Due to the expensive compute cost of retraining a generative model for multiple iterations (c.f. Appendix A.5.4), we plot only one run on each figure. To ensure the reproducibility of our results, we plot the retraining curves for 3 independent runs in Figure 11 in the appendix, illustrating that they have small variance.

**Using probability of one class as reward**. As a first experiment, we filter samples following the probability of the classifier on a predefined class. We arbitrarily chose the class $0$ corresponding to planes. The reward is then defined as $r(x) = \gamma \cdot q_0(x), \gamma > 0$. We plot the evolution of the class proportions as well as the averaged reward across 10 retraining steps in Figure 2 with $\gamma = 5$. Figure 2 shows collapse to the single plane class as the reward increases monotonically, illustrating Lemma 2.2.

**Using the confidence of the classifier as a reward: the emergence of bias**. As a second experiment, we use the confidence of the classifier as a reward, *i.e.,* $r(x) = \gamma \cdot \max_{0 \leq i \leq 9} q_i(x), \gamma > 0$. As written, the reward is therefore uncorrelated from the class but, remains implicitly correlated to it by the fact that *the classifier statistics are class dependent*. In Figure 3 we plot the evolution of the class proportions as well as the average reward. As expected by our theoretical results in Section 2, the average reward increases monotonically. On the other hand, we clearly see that the class proportions become more and more heterogeneous throughout the retraining loop. While confirming our theoretical study this plot therefore additionally shows that retraining on filtered samples increases bias, in a setting where the reward is implicitly correlated to diversity. Taking a step back, this has strong societal and ethical implications as it may imply that in a filtered internet biases may emerge or strengthen as we explain in Section 6.

**Reusing real samples: stability and reward augmentation**. Finally, we illustrate our results from Section 2.2.1 by mixing real and filtered synthetic samples with hyperparameter $\lambda = \frac{1}{2}$. Figure 3 shows that the process remains stable as the proportion of classes remains approximately uniform

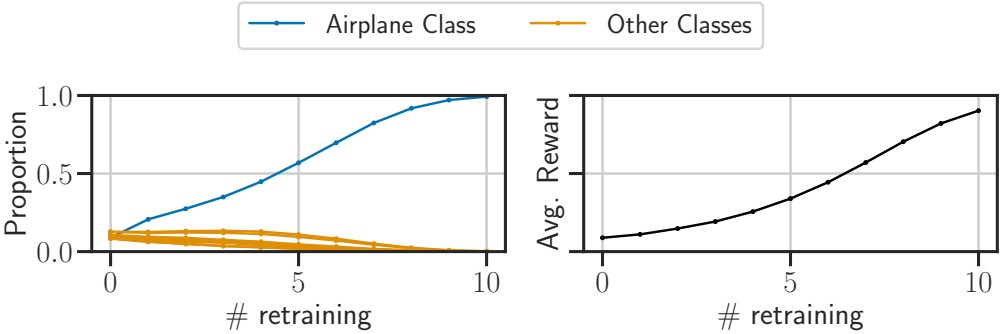

Figure 2: **CIFAR-10**. Evolution of the proportion of the class 'Airplane' and of the 9 other classes when filtering on curated synthetic samples with reward $r(x) = \gamma \cdot q_0(x)$

(as suggested by Theorem 2.3). On the other hand, the average reward increases before stabilizing as predicted by Theorem 2.3.

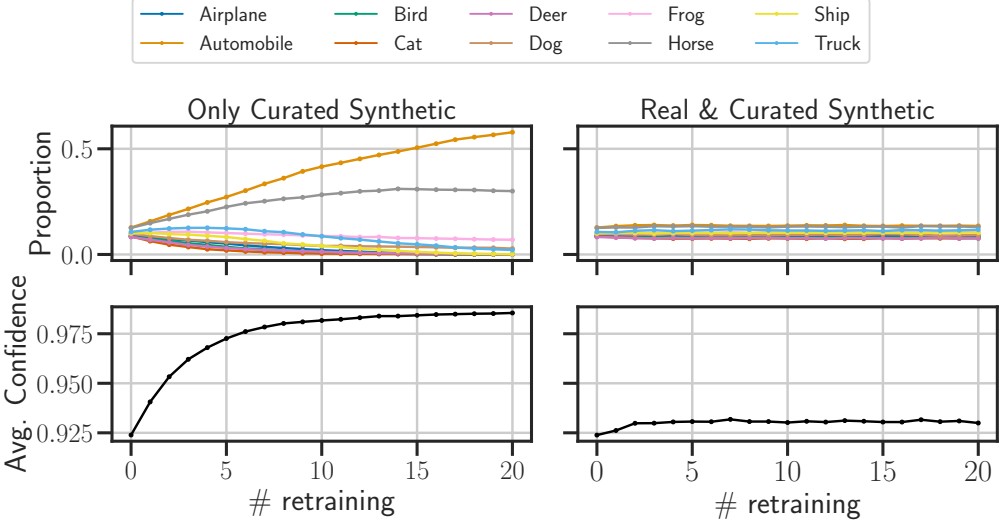

Figure 3: **CIFAR-10**. Evolution of the proportion of each class and the average reward $r(x)$ when filtering based on the confidence of a classifier. On the left, retraining is done solely on the curated synthetic samples which results in the emergence of proportion biases. On the right, retraining is performed on a mixture of real and curated synthetic samples which results in both increased stability and still reward augmentation.

## 5    Conclusion and open questions

We study the impact of data curation on the training of generative models in the self-consuming loop. We provide theoretical results demonstrating that the expected reward underlying the curation process increases and its variance collapses (Lemma 2.2) as well as a convergence result (Theorem 2.1) for the generative model. We additionally provide stability guarantees when reusing real data at each step (Theorem 2.3 and Theorem 2.4) establishing close connections with RLHF and preference optimization. Our work sheds light and theoretically grounds a novel phenomenon: increasing the proportion of curated synthetic data on the Web automatically optimizes preferences for future trained large models. A limitation is that we do not propose an algorithm to address emerging issues like bias amplification as we feel it goes beyond the scope of our paper and is a substantially complex field already intensively explored (Grover et al., 2019; Wyllie et al., 2024; Chen et al., 2024b). We believe, however, that it should be a research priority and constitutes an interesting

avenue for future work. Another interesting direction is to study the impact of the particular reward function underlying filtering (confidence, quality, diversity...) on the emerging bias amplification.

## 6 Broader impacts

Training and aligning large generative models are prone to substantial ethical concerns regarding their alignment objective (Shen et al., 2023), representational disparities of the training datasets (Clemmensen and Kjærsgaard, 2022), or the presence of harmful images in the datasets (Birhane et al., 2021; Schramowski et al., 2023; Birhane et al., 2024). Our work mostly focuses on the impact of the curation of synthetic datasets which itself heavily depends on the agent performing the curation and its underlying reward function. In particular the documentation of the Simulacra Aesthetic Captions dataset (Pressman et al., 2022) alerts that the human-based curation step is performed by a group of individuals that lacks diversity, mostly from Western, Educated, Industrialized, Rich, and Democratic (WEIRD) individuals (Henrich et al., 2010). A similar bias is likely occurring in the JourneyDB (Pan et al., 2023) dataset and, more generally, in the synthetic data appearing on the web. However, our work mostly revolves around a theoretical analysis and raises awareness of the implicit alignment and potential bias amplification of self-consuming generative models. We therefore firmly believe that the potential benefits of this awareness outweigh the potential unforeseen negative consequences of this work.

## 7 Acknowledgements

We sincerely thank Sophie Xhonneux for providing feedback and corrections on the paper. We further thank Mats L. Richter and William Buchwalter for fruitful discussions about the experiments. We thank Mila Cluster for access to computing resources, especially GPUs. AJB is partially funded by an NSERC Post-doc fellowship and an EPSRC Turing AI World-Leading Research Fellowship.

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

# A    Appendix / supplemental material

## A.1    Extended related work

**On retraining with data curation to align on preferences**. In Gupta and Zou (2019), the authors tackle the problem of generating synthetic DNA sequences using Generative Adversarial Networks (GAN, Goodfellow et al. (2014)). They introduce an external function analyzer to rate synthetic samples from the generator and add the highest-scored ones into the discriminator training set. In Yao et al. (2022), the authors propose a new GAN framework to incorporate users preferences in the training. They show state of the art results in generating the user-desired data distribution and theoretically prove the convergence of their method. The key difference to our work is that they aim to generate a diversity of samples that are desired by users and their focus is therefore not on the collapse of their method to a maximal reward set.

**On the impact of alignment on diversity**. In Kirk et al. (2023), the authors investigate how the different stages of alignment affect a models generalization capabilities and output diversity. They empirically show that the output diversity of the RLHF policy is decreased w.r.t. the supervised finetuned policy, which is consistent with our theoretical insights (e.g. lem. 2.2, thm 2.1). However there are major differences with our setting as their contribution is empirical and we investigate the impact of iteratively retraining a model several times on synthetic samples while they study a single training round using RLHF. In Perez et al. (2022), the authors investigate how a language model can be used to generate prompts that lead to a harmful behavior of another language model (red teaming). This is useful to prevent such behavior before public deployment of a large language model. They find that LM-red teaming is a powerful tool that successfully unveils harmful behaviors of language models and help mitigate them. However, such a method can suffer from a diversity problem in the generated prompts if the red-teaming language model is itself biased. In Gao et al. (2023), the authors study how over-optimizing an imperfect proxy of a reward model affects the average reward of the ground-truth reward. They uncover for both reinforcement learning based and best-of-n methods, functional forms of the underlying reward model scores as a function of the KL between the optimized policy and the initial policy that they confirm empirically. In Bai et al. (2022), the authors propose to use iterative online learning for RLHF, where the preferences are updated on a weekly basis using human feedback. In particular at each iteration step they use their best RLHF policy to produce comparisons submitted to crowd-workers. The comparisons are then mixed with existing data and the process continues. The intuition for such methods is that diversity could be improved compared to RLHF since the dataset is generated using different states of the reward models. They additionally empirically show benefits of this practice on metrics such as Elo scores as evaluated by crowd-workers.

**On model collapse**. Briesch et al. (2023) investigates experimentally the self-consuming loop specifically in the case of LLMs and evidences model collapse in that setting. Feng et al. (2024) show that pruning unwanted samples or selecting the best ones from multiple synthetic samples can prevent model collapse in the self-consuming loop.

**Iterative finetuning of language models and rejection-sampling**. While there is already a large literature on RLHF to iteratively finetune LLMs, rejection-sampling is one way to optionally see finetuning as a sampling problem amenable to probabilistic inference. Indeed, recent works Zhao et al. (2024); Kong et al. (2024) frame iterative finetuning as drawing samplesusing rejection sampling, Twisted Sequential Monte Carlo etcfrom the unnormalized posterior distribution $p(x) \propto e^{r(x)}p_0(x)$,where $p_0(x)$ is an initial generative model trained on real data. From this perspective, our framework studies the case where we curate data by using human reward and obtain $x \sim p(x)$ which are samples from the posterior, without access to the density. This allows us to then finetune $p_0(x)$ to approximate the posterior $p(x)$, which in our notation is a step of iteratively finetuning on curated data.

**Accounting for the accumulation of data**. In Gerstgrasser et al. (2024), the authors show that model collapse is evitable, in the case where a model is retrained on the accumulation of all its previous iteraions (and not only data generated from the last iteration). We believe that their setting could be adapted in order to show that accumulating data provides additional stability to the retraining loop and avoids collapse of the reward variance. We leave this exciting research direction as future work.

## A.2 Extension to using a mixture of rewards

Frameworks going beyond a single reward model are especially relevant in practical scenarios in LLM alignment. An interesting reference on this topic is the recent work Munos et al. (2023) which addresses such extension by learning a preference model of samples given a prompt $P(x \succ x|y)$ (as a function of the two variables $x, x$)— instead of the Plackett-Luce reward model $r(x)$ (less general when preferences are non-transitive) which they refer to as Nash Learning from Human Feedback.

We now outline how to derive an extension to a mixture of reward to our setting: First, we can introduce a new latent variable $z$ that describes the randomness in the reward used, which leads to the following expression of the curated distribution after one step of curation:

$$p_{t+1}(x) = p_t(x) \cdot H_{p_t}^K(x) \quad \text{with} \quad H_{p_t}^K(x) := \mathbb{E}_{x_1,\ldots,x_{K-1}\sim p_t,u} \left[ \frac{K \cdot e^{r(x;u)}}{e^{r(x;u)} + \sum_{i=1}^{K-1} e^{r(x_i;u)}} \right]$$

In our setting, we were able to prove that the expected reward increases and the distribution converges to the maximum level set of a unique reward (Lem 2.2, Thm 2.1). However, in the presence of multiple rewards, it is not straightforward that the rewards have the same maximal level sets. Therefore this may yield interesting dynamics and the convergence of $p_t$ may differ. We believe such an extension of our results is outside of the scope of this work and think that it is a fascinating avenue for future work. For example, it may be interesting to study if a reward component in the mixture dominates, thereby dictating the convergence, e.g. if it gets large differences between two samples. In that case, the distribution may converge to only one maximal level set introducing a new model collapse behavior as the mixture of rewards would be dictated by a single reward.

## A.3 Comparison of our results against other results from the litterature

### A.3.1 On retraining from scratch vs iterative fine-tuning

**1. Experiments**. All retraining step in experiments on mixture of Gaussians and two moons are performed from scratch, whereas in the case of CIFAR dataset, due to the high compute cost of retraining the model from scratch (20 hours on an A100 GPU) we performed fine-tuning at each step. Fine-tuning is always performed on $10^6$ images which corresponds to 20 epochs on the original dataset using batch size 128 (i.e. $7.8 * 10^3$ gradient steps). We always use the same amount of images for fair comparison between different proportions of real data injected. In contrast, in Alemohammad et al. (2024), the collapse is shown when the model is retrained from scratch at each iteration. In Shumailov et al. (2023), the experiments are performed using retraining from scratch for Variational AutoEncoders and Gaussian Mixture Models and sequential fine-tuning for Large Language Models. In Bertrand et al. (2024), toy experiments on two moons and mixture of Gaussians are performed by retraining from scratch while experiments on CIFAR10, FFHQ are performed using iterative fine-tuning. It is worth noting that stability using real data in the setting of iterative fine-tuning is easier to obtain than when retraining from scratch, since the model parameters are initialized around a good potential set of parameters. However, we point out that model collapse occurs also in the setting of iterative fine-tuning as shown in Figure 2 of Bertrand et al. (2024) (red curves).

**2. Theory**. Finally regarding our theoretical results, only theorem 2.2 happens in the setting of iterative fine-tuning (since it uses the same setting as in Bertrand et al. (2024)). However, all our other results and in particular theorem 2.1, 2.3, 2.4 do not explicitly assume a special learning algorithm in parameter space. Instead, we consider having a perfect learning model and consider the evolution of the expected reward for such a learning model. In that sense, it applies to learning from scratch with the additional assumption that the model attained perfectly fits the curated distribution.

### A.3.2 On fresh real vs fixed real data

Alemohammad et al. (2024) studies the self-consuming loop in three different settings where the model is retrained a) only on synthetic data b) on a mixture of synthetic data and a fixed set of real data samples b) on synthetic data and a fresh set of real data samples at each step. In setting (a), the retraining loop collapses. In (b) it collapses too but with some delay related to the amount of fixed real data. In (c) the retraining loop does not degrade performances provided there is enough fresh real data at each step. Bertrand et al. (2024) proved stability in the setting (b) under some theoretical

assumptions and in the iterative finetuning framework. Comparatively, our experiments on mixtures of Gaussians are performed using fresh real data at each step while the CIFAR experiments are performed in the fixed real data framework.

## A.4 Proofs

### A.4.1 Proof of Lemma 2.1

**Lemma 2.1.** *Let $p_{t+1}$ be defined as in Equation 4. If $\mathcal{P} = \mathcal{P}(\mathbb{R}^d)$ is the set of probability distributions on $\mathbb{R}^d$, and if we assume that $\mathbb{E}_{y \sim p_t}\left[e^{r(y)}\right] < \infty$, then we have for all $x \in \mathbb{R}^d$,*

$$p_{t+1}(x) \xrightarrow{K \to \infty} p_t(x)\frac{e^{r(x)}}{\mathbb{E}_{\tilde{x} \sim p_t}\left[e^{r(\tilde{x})}\right]} \ . \tag{5}$$

*Proof.* First, by minimization of the cross-entropy, we know that for any distribution $q$, $\arg\max_p \mathbb{E}_{x \sim q}[\log(p(x))] = q$. Therefore, if $p_{t+1}$ is the solution of Equation 4, then we have directly that $p_{t+1}$ has the law of $\hat{x}$, where $\hat{x}$ is defined in Equations 1 and 2. We can now specify explicitly the distribution $p_{t+1}$. Let $p_t$ be the current distribution at time $t$. We first sample $x_1, \cdots, x_K \overset{i.i.d.}{\sim} p_t$. and then independently sample an index $i_K$ following the Plackett-Luce model:

$$\mathbb{P}(i_K = i | x_1, \cdots, x_K) = \frac{e^{r(x_i)}}{\sum_{k=1}^K e^{r(x_j)}}. \tag{9}$$

By noting that the events $\{i_K = i\}_{i=1}^K$ are disjoint, we can write the resulting density:

$$p_{t+1}(x) = \sum_{i=1}^K \int_{y_j, j \neq i} p_t(y_1, \cdots, y_{i-1}, x, y_{i+1}, \cdots, y_K)\mathbb{P}(i_K = i | x, y_j, j \neq i) \prod_{j \neq i} dy_j.$$

By independence since the $K$ samples are drawn i.i.d. and since the Plackett-Luce formula is symmetric, all $K$ terms in the sum are equal. This leads to rewriting:

$$p_{t+1}(x) = K \int_{y_1, \cdots, y_{K-1}} p_t(y_1, \cdots, y_{K-1}, x)\mathbb{P}(i_K = K | y_1, \cdots, y_{K-1}, x) dy_1 \cdots dy_{K-1}$$

$$= p_t(x)K \int_{y_1, \cdots, y_{K-1}} \frac{e^{r(x)}}{e^{r(x)} + \sum_{i=1}^{K-1} e^{r(y_i)}} p_t(y_1) \cdots p_t(y_{K-1}) dy_1 \cdots dy_{K-1}$$

$$= p_t(x) \cdot H_{p_t}^K(x)$$

where

$$H_{p_t}^K(x) = \int_{y_1, \cdots, y_{K-1}} \frac{e^{r(x)}}{\frac{e^{r(x)}}{K} + \sum_{i=1}^{K-1} \frac{e^{r(y_i)}}{K}} p_t(y_1) \cdots p_t(y_{K-1}) dy_1 \cdots dy_{K-1} \tag{10}$$

We now can study the limit $K \to \infty$. Consider the random variable $X = e^{r(x)}$ as $x \sim p_t$. By assumption, $\mathbb{E}[X] < \infty$. We can therefore apply the law of large numbers. Namely, if $X_1, \cdots, X_{K-1}$ are sampled i.i.d.:

$$\frac{1}{K-1}(X_1 + \cdots + X_{K-1}) \xrightarrow{\mathbb{P}} \mathbb{E}[X] \tag{11}$$

Furthermore, for all $x, y_1, \ldots, y_{K-1}$, we have $0 \leq \frac{e^{r(x)}}{e^{r(x)} + \sum_{i=1}^{K-1} e^{r(y_i)}} \leq 1$ and $\frac{e^{r(x)}}{K} \xrightarrow{K \to \infty} 0$.

Rewriting Equation 10:

$$H_{p_t}^K(x) = \int_{y_1, \cdots, y_{K-1}} \frac{e^{r(x)}}{\frac{e^{r(x)}}{K} + \frac{K-1}{K} \frac{\sum_{i=1}^{K-1} e^{r(y_i)}}{K-1}} p_t(y_1) \cdots p_t(y_{K-1}) dy_1 \cdots dy_{K-1}$$

we get that:

$$H_{p_t}^K(x) \xrightarrow{K \to \infty} \frac{e^{r(x)}}{\mathbb{E}_{y \sim p_t}\left[e^{r(y)}\right]}$$

which directly implies

$$p_{t+1}(x) \xrightarrow{K \to \infty} p_t(x)\frac{e^{r(x)}}{\mathbb{E}_{y \sim p_t}\left[e^{r(y)}\right]}$$

$\square$

### A.4.2 Additional lemma: the reward expectation is increasing

Without assuming that the reward is bounded, we can show using Jensen inequality that the reward expectation increases at each retraining step.

**Lemma A.1.** *When performing $K$-wise filtering, the expected reward increases, i.e., $\forall t \geq 0$:*

$$\mathbb{E}_{p_{t+1}}\left[e^{r(x)}\right] \geq \mathbb{E}_{p_t}\left[e^{r(x)}\right] \quad . \tag{12}$$

*Proof.* Consider the random variable $Y = \frac{K-1}{K}\frac{\sum_{i=1}^{K-1} e^{r(y_i)}}{K-1}$ when $y_1, \cdots, y_{K-1} \overset{\text{i.i.d.}}{\sim} p_t$.

For $a, b > 0$, the function $x \mapsto \frac{a}{b+x}$ is convex on $\mathbb{R}_+^*$. Hence by Jensen inequality, for any $x$:

$$H_{p_t}^K(x) = \mathbb{E}_Y\left[\frac{e^{r(x)}}{\frac{e^{r(x)}}{K} + Y}\right] \geq \frac{e^{r(x)}}{\frac{e^{r(x)}}{K} + \mathbb{E}[Y]} = \frac{e^{r(x)}}{\frac{e^{r(x)}}{K} + \frac{K-1}{K}\mathbb{E}_{p_t}\left[e^{r(x)}\right]}$$

Finally, we can write:

$$\begin{aligned}
\mathbb{E}_{p_{t+1}}\left[e^{r(x)}\right] &= \int e^{r(x)} p_t(x) H_{p_t}^K(x) dx \\
&\geq \int p_t(x) \frac{e^{2r(x)}}{\frac{e^{r(x)}}{K} + \frac{K-1}{K}\mathbb{E}_{y \sim p_t}\left[e^{r(y)}\right]} dx \\
&\geq \frac{\mathbb{E}_{x \sim p_t}\left[e^{r(x)}\right]^2}{\frac{\mathbb{E}_{x \sim p_t}\left[e^{r(x)}\right]}{K} + \frac{K-1}{K}\mathbb{E}_{y \sim p_t}\left[e^{r(y)}\right]} \\
&= \mathbb{E}_{x \sim p_t}\left[e^{r(x)}\right]
\end{aligned}$$

where we have used again Jensen inequality on the convex function $\frac{x^2}{\frac{x}{K}+c}$ on $\mathbb{R}_+^*$ where

$$c := \frac{K-1}{K}\mathbb{E}_{y \sim p_t}\left[e^{r(y)}\right] > 0$$

$\square$

### A.4.3 Proof of Lemma 2.2

**Lemma 2.2.** *Let $p_{t+1}$ be the distribution induced from a discrete choice model on $p_t$ (Equation 4). Suppose Assumption 2.1 B holds, then the expected reward increases proportionally to its variance at each retraining iteration:*

$$\mathbb{E}_{p_{t+1}}\left[e^{r(x)}\right] \geq \mathbb{E}_{p_t}\left[e^{r(x)}\right] + \frac{K-1}{K}\frac{\text{Var}_{p_t}\left[e^{r(x)}\right]}{e^{r*}} \quad . \tag{7}$$

*Especially the expected reward converges to the maximum reward and its variance vanishes:*

$$\mathbb{E}_{p_t}\left[e^{r(x)}\right] \xrightarrow{t\to\infty} e^{r_*} \quad \text{and} \quad \text{Var}_{p_t}\left[e^{r(x)}\right] \xrightarrow{t\to\infty} 0 \ .$$

*Proof.* By symmetry, we can write:

$$K\mathbb{E}_{p_{t+1}}\left[e^{r(x)}\right] = \int_{x_1,\cdots,x_K} K\frac{e^{2r(x_1)}+\cdots+e^{2r(x_K)}}{e^{r(x_1)}+\cdots+e^{r(x_K)}}\prod_{k=1}^{K}p_t(x_k)dx_k$$

$$= \int_{x_1,\ldots,x_K}\sum_{j=1}^{K}\left[e^{r(x_j)}\frac{e^{r(x_1)}+\cdots+e^{r(x_K)}}{e^{r(x_1)}+\cdots+e^{r(x_K)}}+e^{r(x_j)}\frac{(K-1)e^{r(x_j)}-\sum_{i\neq j}e^{r(x_i)}}{e^{r(x_1)}+\cdots+e^{r(x_K)}}\right]$$

$$\prod_{k=1}^{K}p_t(x_k)dx_k$$

$$= K\mathbb{E}_{p_t}\left[e^{r(x)}\right] + \int_{x_1,\ldots,x_K}\frac{\sum_{i<j}\left(e^{r(x_i)}-e^{r(x_j)}\right)^2}{e^{r(x_1)}+\cdots+e^{r(x_K)}}\prod_{k=1}^{K}p_t(x_k)dx_k$$

$$\leq K\mathbb{E}_{p_t}\left[e^{r(x)}\right] + \sum_{i<j}\frac{2\text{Var}_{p_t}\left[e^{r(x)}\right]}{Ke^{r_*}}$$

$$\leq K\mathbb{E}_{p_t}\left[e^{r(x)}\right] + \frac{K(K-1)}{2}\frac{2\text{Var}_{p_t}\left[e^{r(x)}\right]}{Ke^{r_*}}$$

$$\leq K\mathbb{E}_{p_t}\left[e^{r(x)}\right] + \frac{(K-1)\text{Var}_{p_t}\left[e^{r(x)}\right]}{e^{r_*}}$$

This brings finally,

$$\mathbb{E}_{p_{t+1}}\left[e^{r(x)}\right] \geq \mathbb{E}_{p_{t+1}}\left[e^{r(x)}\right] + \frac{K-1}{K}\frac{\text{Var}_{p_t}\left[e^{r(x)}\right]}{e^{r_*}}$$

We now prove that the expected reward converges and we will first show the following lemma:

**Lemma A.2.** $\forall\varepsilon\geq 0, \forall t\geq 0,$
$$\mathbb{P}_{t+1}(r(x)\geq r_*-\varepsilon)\geq\mathbb{P}_t(r(x)\geq r_*-\varepsilon) \tag{13}$$

*Proof.* Consider $(x_1,\ldots,x_K)\overset{i.i.d.}{\sim} p_t$ and denote $\mathcal{B}_\varepsilon := \{x, r(x)\geq r_*-\varepsilon\}$. Then,

$$\mathbb{P}_t(r(x)\geq r_*-\varepsilon) = \frac{1}{K}\mathbb{E}_{x_1,\ldots,x_K}\left[\sum_{i=1}^{K}\mathbb{1}_{x_i\in\mathcal{B}_\varepsilon}\right] \ .$$

On the other hand,

$$\mathbb{P}_{t+1}(r(x)\geq r_*-\varepsilon) = \mathbb{E}_{x_1,\ldots,x_K}\left[\sum_{i=1}^{K}\mathbb{1}_{x_i\in\mathcal{B}_\varepsilon}\frac{e^{r(x_i)}}{\sum_{k=1}^{K}e^{r(x_k)}}\right]$$

Proving [Lemma A.2](#) is then equivalent, by permutation symmetries to showing that $\forall k\leq K$, if $r(x_1),\ldots,r(x_k)\geq r_*-\varepsilon$ and $r(x_{k+1}),\ldots,r(x_K)<r_*-\varepsilon$, then $\frac{k}{K}\leq\sum_{i=1}^{k}\frac{e^{r(x_i)}}{\sum_{k=1}^{K}e^{r(x_k)}}$.

We can then write:

$$\sum_{i=1}^{k} \frac{e^{r(x_i)}}{\sum_{k=1}^{K} e^{r(x_k)}} = \frac{\sum_{i=1}^{k} e^{r(x_i)}}{\sum_{k=1}^{K} e^{r(x_k)}}$$

$$= \frac{k\mu_1}{k\mu_1 + (K-k)\mu_2}$$

$$\geq \frac{k}{K}$$

Where $\mu_2 := \frac{\sum_{i=k+1}^{K} e^{r(x_i)}}{K-k} \leq \frac{\sum_{i=1}^{k} e^{r(x_i)}}{k} =: \mu_1$ □

Let $\varepsilon > 0$. By assumption on $r_*$(Assumption 2.1 B), we know that there exists $\delta > 0$ such that $\mathbb{P}_0(r(x) \geq r_* - \varepsilon) \geq \delta$ and hence using Lemma A.2, $\forall t \geq 0, \mathbb{P}_t(r(x) \geq r_* - \varepsilon) \geq \delta$. Therefore, while $\mathbb{E}_{p_t}\left[e^{r(x)}\right] \leq e^{r_*} - 2\varepsilon$, we know that

$$\mathrm{Var}_{p_t}\left[e^{r(x)}\right] \geq \varepsilon^2 \mathbb{P}_t(r(x) \geq r_* - \varepsilon) \geq \varepsilon^2 \delta \ .$$

Therefore, while $\mathbb{E}_{p_t}\left[e^{r(x)}\right] \leq e^{r_*} - 2\varepsilon$, we have using Lemma 2.2 that

$$E_{p_{t+1}}\left[e^{r(x)}\right] \geq \mathbb{E}_{p_{t+1}}\left[e^{r(x)}\right] + \frac{K-1}{K} \frac{\varepsilon^2 \delta}{e^{r_*}} \ .$$

Since $\frac{K-1}{K} \frac{\varepsilon^2 \delta}{e^{r_*}} > 0$, this can happen for only a finite number of steps and hence we know that there exists a time $T_\varepsilon \geq 0$ such that (remind that the expectation of the reward is increasing by Lemma 2.2):

$$\forall t \geq T_\varepsilon, \quad \mathbb{E}_{p_t}\left[e^{r(x)}\right] > e^{r_*} - 2\varepsilon \ .$$

Since, the expected reward is obviously recursively bounded by $e^{r_*}$ at any iteration $t$, we just have proved that it converges.

We now prove that the variance has finite sum. Indeed, just notice that using Lemma 2.2 that $\forall T \geq 0$:

$$\sum_{t=0}^{T} \mathrm{Var}_{p_t}\left[e^{r(x)}\right] \leq e^{r_*} \frac{K}{K-1} \left( \mathbb{E}_{p_{T+1}}\left[e^{r(x)}\right] - \mathbb{E}_{p_0}\left[e^{r(x)}\right] \right)$$

$$\leq \frac{K}{K-1} e^{2r_*} \ .$$

This proves that $\sum_{t=0}^{T} \mathrm{Var}_{p_t}\left[e^{r(x)}\right] < \infty$. Especially since the reward variance has finite sum and is positive, it converges to 0. □

### A.4.4 Fixed points of the retraining loop with filtering

**Lemma A.3.** *A probability density $p$ is a fixed point of Equation 10 if and only if it puts all its mass on a single level set of the reward function. In other words, there exists $r_* \in \mathbb{R}$ such that $\mathbb{P}(r(x) = r_*) = 1$.*

*Proof.* Given the density $p$, denote $\mathbb{P}$ the corresponding probability function and $\mathcal{F}^K(p)$ the curated distribution using Equations 1 and 2. When the reward $r$ is $p$-a.s. bounded, this is a direct consequence of Lemma 2.2. When this is not the case, we know the existence of two disjoint interval $I, J \subset \mathbb{R}$ such that $\mathbb{P}(r(x) \in I) > 0$ and $\mathbb{P}(r(x) \in J) > 0$. From the proof of Lemma 2.2, we have seen that, taking $p_t := p$:

$$K\mathbb{E}_{\mathcal{F}^K(p)}\left[e^{r(x)}\right] = K\mathbb{E}_p\left[e^{r(x)}\right] + \int_{x_1,\dots,x_K} \frac{\sum_{i<j}\left(e^{r(x_i)} - e^{r(x_j)}\right)^2}{e^{r(x_1)} + \cdots + e^{r(x_K)}} \prod_{k=1}^{K} p(x_k) dx_k$$

$$> K\mathbb{E}_p\left[e^{r(x)}\right]$$

using that $I, J$ have strictly positive mass and disjoint rewards. Therefore, $p$ cannot be a fixed point.

Conversely, if $p$ puts mass on a single level set of $r$, it is straightforward that it is a fixed point of the filtering operator because $H_p^K(x)$ is almost surely constant. □

### A.4.5 Proof of Theorem 2.1

**Theorem 2.1.** *Let for all $t \geq 0$, $p_{t+1}$ be the distribution induced from a discrete choice model on $p_t$ (Equation 4) where $\mathcal{P} = \mathcal{P}(\mathbb{R}^d)$ is the set of probability distributions on $\mathbb{R}^d$. If $p_0$ satisfies Assumption 2.1 C, then we can define $p_*(x) := \frac{p_0(x)\mathbb{1}_{r(x)=r_*}}{\mathbb{P}_0(r(x)=r_*)}$ and the self-consuming loop on curated samples $p_t$ converges to $p_*$:*

$$D_{\mathrm{KL}}(p_*||p_t) \xrightarrow{t\to\infty} 0.$$

*Proof.* Recall $p_*(x) = \frac{p_0(x)\mathbb{1}_{r(x)=r_*}}{\mathbb{P}_0(r(x)=r_*)}$. Furthermore, notice that for any $t \geq 0$,

$$p_{t+1}(x)\mathbb{1}_{r(x)=r_*} \propto p_0(x)\mathbb{1}_{r(x)=r_*}$$

by recursion because $H_{p_t}^K(x)$ depends only on $r(x)$. From that we deduce:

$$D_{\mathrm{KL}}(p_*||p_t) = -\log(\mathbb{P}_t(r(x) = r_*)).$$

We therefore only have to show that $\mathbb{P}_t(r(x) = r_*) \xrightarrow{t\to\infty} 1$.

We will first show the following lemma:

---

**Lemma A.4.** $\forall \varepsilon \geq 0, \forall t \geq 0$,

$$\mathbb{P}_{t+1}(r(x) = r_*) - \mathbb{P}_t(r(x) = r_*) \geq \mathbb{P}_0(r(x) = r_*) * (\mathbb{P}_{t+1}(r(x) \geq r_* - \varepsilon) - \mathbb{P}_t(r(x) \geq r_* - \varepsilon))$$

---

*Proof.* We will actually show:

$$\mathbb{P}_{t+1}(r(x) = r_*) - \mathbb{P}_t(r(x) = r_*) \geq \mathbb{P}_t(r(x) = r_*) * (\mathbb{P}_{t+1}(r(x) \geq r_* - \varepsilon) - \mathbb{P}_t(r(x) \geq r_* - \varepsilon)) \tag{14}$$

from what we directly deduce Lemma A.4 by using Lemma A.2.

To prove Equation 14, just notice that for any $x, y$, if $r(x) \geq r(y)$ then $H_{p_t}^K(x) \geq H_{p_t}^K(y)$ by increasing monotonicity of $z \mapsto \frac{z}{z+c}$ on $\mathbb{R}_+^*$ for a positive constant $c > 0$. Therefore we know the existence of a constant $C$ such that $\forall x, y$, if $r(x) = r_*$ and $r(y) \leq r_*$, then $H_{p_t}^K(x) \geq C \geq H_{p_t}^K(y)$. For example, take $C = \inf_{x \text{ s.t. } r(x)=r_*} H_{p_t}^K(x)$. Then we can write:

$$
\begin{aligned}
\mathbb{P}_{t+1}(r(x) = r_*) - \mathbb{P}_t(r(x) = r_*) &= \int \mathbb{1}_{r(x)=r_*}(p_{t+1}(x) - p_t(x))dx \\
&= \int \mathbb{1}_{r(x)=r_*}p_t(x)(H_{p_t}^K(x) - 1)dx \\
&\geq \int \mathbb{1}_{r(x)=r_*}p_t(x)(C - 1)dx \\
&= \mathbb{P}_t(r(x) = r_*)(C - 1)
\end{aligned}
$$

and:

$$
\begin{aligned}
\mathbb{P}_{t+1}(r(x) \geq r_* - \varepsilon) - \mathbb{P}_t(r(x) \geq r_* - \varepsilon) &= \int \mathbb{1}_{r(x)\geq r_*-\varepsilon}(p_{t+1}(x) - p_t(x))dx \\
&= \int \mathbb{1}_{r(x)\geq r_*-\varepsilon}(H_{p_t}^K(x) - 1)dx \\
&\leq \int \mathbb{1}_{r(x)\geq r_*-\varepsilon}p_t(x)(C - 1)dx \\
&= \mathbb{P}_t(r(x) \geq r_* - \varepsilon)(C - 1) \\
&\leq (C - 1)
\end{aligned}
$$

where in the last step we have used $C - 1 \geq 0$ because $\mathbb{P}_{t+1}(r(x) \geq r_* - \varepsilon) - \mathbb{P}_t(r(x) \geq r_* - \varepsilon) \geq 0$ by Lemma A.2 and $\mathbb{P}_t(r(x) \geq r_* - \varepsilon) \leq 1$.

Combining the last two equations we get:

$$\mathbb{P}_{t+1}(r(x) = r_*) - \mathbb{P}_t(r(x) = r_*) \geq \mathbb{P}_t(r(x) = r_*) * (\mathbb{P}_{t+1}(r(x) \geq r_* - \varepsilon) - \mathbb{P}_t(r(x) \geq r_* - \varepsilon))$$

$\square$

We can now prove $\mathbb{P}_t(r(x) = r_*) \xrightarrow{t \to \infty} 1$. Let $\delta > 0$, suppose that at time $t$,
$$\mathbb{P}_t(r(x) = r_*) \leq 1 - \delta \ .$$
Denote for $\varepsilon > 0$, $\mathcal{U}_\varepsilon = \{x \in \mathbb{R}^d | r_* > r(x) \geq r_* - \varepsilon\}$. We know that $\bigcap_{\varepsilon > 0} \mathcal{U}_\varepsilon = \varnothing$. Therefore, $\exists \varepsilon^t > 0$ such that $\mathbb{P}_t(\mathcal{U}_{\varepsilon^t}) \leq \frac{\delta}{4}$. Furthermore, for any $t' \geq t$, we know that

$$\mathbb{P}_{t'}(r(x) \leq r_* - \varepsilon^t) \xrightarrow{t' \to \infty} 1 \tag{15}$$

by convergence of the expectation (Lemma 2.2) and Markov property. We therefore know that $\exists t' \geq t$ such that $\mathbb{P}_{t'}(r(x) \leq r_* - \varepsilon^t) \geq 1 - \frac{\delta}{2}$.

By using the preceding Lemma A.4, we get:

$$\mathbb{P}_{t'}(r(x) = r_*) - \mathbb{P}_t(r(x) = r_*) \geq p_0 \cdot (\mathbb{P}_{t'}(r(x) \geq r_* - \varepsilon^t) - \mathbb{P}_t(r(x) \geq r_* - \varepsilon^t))$$
$$\geq \mathbb{P}_0(r(x) = r_*) \cdot ((1 - \frac{\delta}{2}) - (1 - \delta + \frac{\delta}{4}))$$
$$\geq \mathbb{P}_0(r(x) = r_*) \cdot \frac{\delta}{4}$$

and $\mathbb{P}_t(r(x) = r_*)$ hence increases by at least $\frac{\delta}{4}$. Therefore, the condition $\mathbb{P}_t(r(x) = r_*) \leq 1 - \delta$ must become invalid at some point. Since we have shown this for any $\delta > 0$, this shows that $\mathbb{P}_t(r(x) = r_*) \to 1$.

$\square$

### A.4.6 Proof of Theorem 2.3

**Theorem 2.3.** *Let $\lambda > 0$ and consider the process $(p_t)$ defined in eq. 8, with $p_0 = p_{\text{ref}}$. If $p_{\text{ref}}$ satisfies Assumption 2.1 B, then for all $t \geq 1$:*

$$\mathbb{E}_{p_t}\left[e^{r(x)}\right] \geq \mathbb{E}_{p_{\text{ref}}}\left[e^{r(x)}\right] + \frac{\lambda}{(1+\lambda)^3} \frac{(K-1)\text{Var}_{p_{\text{ref}}}\left[e^{r(x)}\right]}{K e^{r_*}} \ .$$

*Proof.* We proceed by recursion. First, we know that $\forall t \geq 1, \text{Var}_{p_t}\left[e^{r(x)}\right] \geq \left(\frac{1}{1+\lambda}\right)^2 \text{Var}_{p_{\text{ref}}}\left[e^{r(x)}\right]$. Furthermore it is straightforward using Lemma A.1 and a recursion that $\forall t \geq 0, \mathbb{E}_{p_t}\left[e^{r(x)}\right] \geq \mathbb{E}_{p_{\text{ref}}}\left[e^{r(x)}\right]$.

This brings that

$\forall t \geq 1, \mathbb{E}_{p_t}\left[e^{r(x)}\right] \geq \frac{1}{1+\lambda}\mathbb{E}_{p_{\text{ref}}}\left[e^{r(x)}\right] + \frac{\lambda}{1+\lambda}\mathbb{E}_{p_{\text{ref}}}\left[e^{r(x)}\right] + \frac{\lambda}{(1+\lambda)^3}\frac{(K-1)\text{Var}_{p_{\text{ref}}}\left[e^{r(x)}\right]}{K e^{r_*}}$ which brings the result.

$\square$

We can actually show the following lower bound on the limit:

**Lemma A.5.** *Consider the process $p_{t+1}(x) = \frac{1}{1+\lambda}p_{\text{ref}}(x) + \frac{\lambda}{1+\lambda}p_t(x) \cdot H_{p_t}^K(x)$ with $p_0 = p_{\text{ref}}$. Then,*

$$\liminf_{t \to \infty} \mathbb{E}_{p_t}\left[e^{r(x)}\right] \geq \mathbb{E}_{p_{\text{ref}}}\left[e^{r(x)}\right] + \frac{\lambda}{(1+\lambda)^2}\frac{(K-1)\text{Var}_{p_{\text{ref}}}\left[e^{r(x)}\right]}{K e^{r_*}} \ .$$

*Proof.* Using the proof of Theorem 2.3 we can show the following more precise lower bound at each step: denote $A := \frac{\lambda}{1+\lambda}$ and $B = \frac{1}{(1+\lambda)^2} \frac{(K-1)\text{Var}_{p_{\text{ref}}}\left[e^{r(x)}\right]}{Ke^{r_*}}$, then for all $t \geq 1$:

$$\mathbb{E}_{p_t}\left[e^{r(x)}\right] \geq \mathbb{E}_{p_{\text{ref}}}\left[e^{r(x)}\right] + A^t B + A^{t-1}B + \cdots + AB \ .$$

This directly bring that :

$$\begin{aligned}
\liminf_{t\to\infty}\mathbb{E}_{p_t}\left[e^{r(x)}\right] &\geq \mathbb{E}_{p_{\text{ref}}}\left[e^{r(x)}\right] + AB\sum_{i=0}^{\infty} A \\
&= \mathbb{E}_{p_{\text{ref}}}\left[e^{r(x)}\right] + \frac{AB}{1-A} \\
&= \mathbb{E}_{p_{\text{ref}}}\left[e^{r(x)}\right] + \frac{\lambda}{1+\lambda}\frac{1}{(1+\lambda)^2}\frac{(K-1)\text{Var}_{p_{\text{ref}}}\left[e^{r(x)}\right]}{Ke^{r_*}}\frac{1}{1-\frac{\lambda}{1+\lambda}} \\
&= \mathbb{E}_{p_{\text{ref}}}\left[e^{r(x)}\right] + \frac{\lambda}{(1+\lambda)^2}\frac{(K-1)\text{Var}_{p_{\text{ref}}}\left[e^{r(x)}\right]}{Ke^{r_*}}
\end{aligned}$$

$\square$

### A.4.7 Proof of Theorem 2.4

**Theorem 2.4.** *Let $\lambda > 0$ and $p_{\text{ref}} \in \mathcal{P}(\mathbb{R}^d)$ with a density with respect to Lebesgue measure. Consider the process $(p_t)$ defined in Equation 8, with $p_0 = p_{\text{ref}}$. Suppose that $\lambda < \frac{1}{K-1}$, then, for all $t \geq 1$*

$$D_{\text{KL}}(p_t||p_{\text{ref}}) \leq -\log\left(1 - \lambda(K-1)\right) \ .$$

*Proof.* We know that $\forall K \geq 2, \forall x \in \mathbb{R}^d, H_{p_t}^K(x) \leq K$.

We can then show by recursion that $\forall t \geq 1, \forall x, \frac{p_t(x)}{p_{\text{ref}}(x)} \leq \frac{1}{1-\lambda(K-1)}$. Indeed, it is true at initialization and if true at time $t$, then at time $t+1$:

$$\frac{p_{t+1}(x)}{p_{\text{ref}}(x)} \leq \frac{1}{1+\lambda} + \frac{\lambda}{1+\lambda}\frac{1}{1-\lambda(K-1)} \cdot K \leq \frac{1}{1-\lambda(K-1)}$$

We then just replace this bound in the expression of the $D_{\text{KL}}(p_t||p_{\text{ref}})$:

$$D_{\text{KL}}(p_t||p_{\text{ref}}) = \mathbb{E}_{p_t}\left[\log(\frac{p_t(x)}{p_{\text{ref}}(x)})\right] \leq \log\left(\frac{1}{1-\lambda(K-1)}\right) \ .$$

$\square$

### A.4.8 Additional lemma: retraining on a convex combination of previous iterations

We study here the impact of retraining on a combination of all previous iterations and show that the process remains constant. This motivates and enlightens previous works that consider only retraining on the distribution at the last iteration. Let $\alpha_0, \alpha_1, \alpha_2 \ldots$ a fixed non-negative sequence and consider a retraining process using maximum likelihood: $\theta_{t+1} = \arg\max_\theta \sum_{i=0}^{t} \alpha_i \mathbb{E}_{p_{\theta_i}} \log(p_\theta(x))$. We will assume for this lemma that the solution of this optimization problem is unique. Otherwise the lemma remains valid but for a carefully chosen solution when there are multiple possibilities.

**Lemma A.6.** *Suppose we start with the first $T$ iterations predefined, i.e., by fixing $p_0, \cdots, p_{T-1}$. Then starting $t = T$, the learned distribution is constant, i.e., $\forall t \geq T, p_t = p_T$.*

**Discussion**. As an example, suppose that we take $p_0 = p_{\text{data}}$ and $p_1$ an initial generative model trained on $p_{\text{data}}$. Then, Lemma A.6 states that starting $t = 2$, the learned distribution at each step will be constant equal to $p_2$. In other words, we cannot expect the process to converge to a global maximizer of the data log-likelihood. More generally, Lemma A.6 shows that if the respective proportion of previous iterations remains constant throughout the retraining loop, the process remains constant and hence cannot converge towards the data distribution. These considerations have interesting links with previous work by Gerstgrasser et al. (2024) which experimentally showed that accumulating data with fixed relative ratios breaks the curse of recursion. However, note that the focus is different since they are in the finite sample setting while we study the infinite sample setting. Finally Lemma A.6 implies that to ensure convergence, we need to relatively decrease the proportion of previous iterations and comparatively increase the relative proportion of the data distribution or only use the distribution of the current iteration. This has been done in Bertrand et al. (2024) for parametrized generative models under some assumptions

*Proof.* We prove the result by recursion starting $t = T$. By definition:

$$\theta_T = \arg\max_\theta \sum_{i=0}^{T-1} \alpha_i \mathbb{E}_{p_{\theta_i}} \log(p_\theta(x))$$

Then suppose that for all $j$ such that $T \leq j \leq t$, $\theta_j = \theta_T$. Then we can write:

$$\theta_{t+1} = \arg\max_\theta \sum_{i=0}^{t} \alpha_i \log(p_\theta(x))$$

But we know by cross-entropy minimization that

$$\theta_T = \arg\max_\theta \mathbb{E}_{p_{\theta_T}} \log(p_\theta(x)) = \arg\max_\theta \sum_{i=T}^{t} \mathbb{E}_{p_{\theta_i}} \log(p_\theta(x)) \ .$$

Furthermore, by definition,

$$\theta_T = \arg\max_\theta \sum_{i=0}^{T-1} \alpha_i \mathbb{E}_{p_{\theta_i}} \log(p_\theta(x)) \ .$$

In particular it maximizes the sum of both previous terms and hence $\theta_{t+1} = \theta_T$ ▢

### A.4.9 Additional lemma of convergence in parameters

**Lemma A.7.** $\forall \lambda \in \mathbb{R}_+$, *if* $\lambda < \frac{\alpha}{2L\varepsilon}$, *then for* $\theta_0$ *in a neighborhood of* $\theta_*$, *we have the following rate of convergence:*

$$\|\theta_t - \theta_*\| = \tilde{\mathcal{O}}\left(\left(\frac{\lambda(\alpha + \varepsilon L)}{\alpha + \lambda(\alpha - \varepsilon L)}\right)^t\right) \ . \tag{16}$$

*Proof.* We follow the same steps and notations as in Bertrand et al. (2024). The main idea is to get another bound on the operator norm of the Jacobian at $\theta_*$: $\|\mathcal{J}\mathcal{G}(\theta^*)\|$ (their lemma E.1 (iii)). We begin with their intermediate result (lemma E.1 (ii)):

$$\mathcal{J}\mathcal{G}(\theta^*) = (I + \lambda A^{-1}B)^{-1}\lambda A^{-1}B$$

However we will bound this term differently. First note that $\|B - A\| \leq L\varepsilon$.

From this, we deduce by sub-multiplicativity of the matrix norm that:

$$\|A^{-1}B - I\| \le \|A^{-1}\|\|B - A\| \le \frac{L\varepsilon}{\alpha}$$

and by triangular inequality:

$$\|A^{-1}B\| = \|A^{-1}(B - A) + I\| \le \|A^{-1}\|\|B - A\| + 1 \le 1 + \frac{L\varepsilon}{\alpha} \quad .$$

Now we use the triangular inequality again to write:

$$\|\mathcal{J}\mathcal{G}(\theta^*)\| \le \|(I + \lambda A^{-1}B)^{-1}\|\|\lambda A^{-1}B\| \quad .$$

But,

$$
\begin{aligned}
\|(I + \lambda A^{-1}B)^{-1}\| &= \|((I + \lambda I) + \lambda(A^{-1}B - I))^{-1}\| \\
&= \frac{1}{1+\lambda}\|(I + \frac{\lambda}{1+\lambda}(A^{-1}B - I))^{-1}\| \\
&\le \frac{1}{1+\lambda}\frac{1}{1 - \frac{\lambda}{1+\lambda}\|A^{-1}B - I\|} \\
&\le \frac{1}{1+\lambda - \lambda\frac{L\varepsilon}{\alpha}}
\end{aligned}
$$

where we have used that $\frac{L\varepsilon}{\alpha} < 1$. Finally,

$$\|\mathcal{J}\mathcal{G}(\theta^*)\| \le \lambda\frac{1}{1+\lambda - \lambda\frac{L\varepsilon}{\alpha}}(1 + \frac{L\varepsilon}{\alpha})$$

and a sufficient condition for having $\|\mathcal{J}\mathcal{G}(\theta^*)\| < 1$ is

$$\lambda\frac{1}{1+\lambda - \lambda\frac{L\varepsilon}{\alpha}}(1 + \frac{L\varepsilon}{\alpha}) < 1$$

or equivalently,

$$\lambda < \frac{\alpha}{2L\varepsilon} \quad .$$

With this new bound $\lambda < \frac{\alpha}{2L\varepsilon}$ which ensures that the operator norm of the Jacobian is smaller than 1, *i.e.*, $\|\mathcal{J}\mathcal{G}(\theta^*)\| < 1$, we can unroll the remaining steps of their proof to get Equation 16 □

### A.4.10 Proof of Theorem 2.2

**Theorem 2.2.** *Under Assumption 2.2, if $L\varepsilon < \alpha$ and $\lambda < \frac{\alpha}{2L\varepsilon}$, then there exists a neighborhood of the optimal distribution parameters $\theta_*$ such that for any initial parameters $\theta_0$ in that neighborhood, $p_{\theta_t}$ converges to $p_{\theta_*}$ exponentially fast:*

$$D_{\mathrm{KL}}(p_{\theta_*}||p_{\theta_t}) = \tilde{\mathcal{O}}\left(\left(\frac{\lambda(\alpha + \varepsilon L)}{\alpha + \lambda(\alpha - \varepsilon L)}\right)^{2t}\right) \quad .$$

**Algorithm 1** Iterative retraining with curated synthetic data

---

**input :** $\mathcal{D}_{\text{real}} := \{x_i\}_{i=1}^n, \mathcal{A}$ `// True data, learning procedure,`
**param:** $T, \lambda, \beta$ `// Number of retraining iterations, proportion of gen. data, reward`
        `multiplicative factor`
$p_0 = \mathcal{A}(\mathcal{D}_{\text{real}})$ `// Learn generative model on true data`
**for** $t$ *in* $1, \dots, T$ **do**
    **for** $i$ *in* $1, \dots, \lfloor \lambda \cdot n \rfloor$ **do**
        $\tilde{x}_1, \dots, \tilde{x}_K \sim p_{t-1}$ `// Sample K synthetic data points`
        $\tilde{x}_k$ is selected by a user with probability $\frac{e^{r(\tilde{x}_k)}}{\sum_{j=1}^K e^{r(\tilde{x}_j)}}$,    $1 \le k \le K$ . `// Luce's model`
        $\hat{x}_i \leftarrow \tilde{x}_k$
    $\mathcal{D}_{\text{filtered}} = \{\hat{x}_i\}_{i=1}^{\lfloor \lambda \cdot n \rfloor}$ `// New filtered dataset`
    $p_t = \mathcal{A}(\mathcal{D}_{\text{real}} \cup \mathcal{D}_{\text{filtered}})$ `// Generative model is learned on synthetic and true data`
**return** $p_T$

---

*Proof.* We know that $\theta_*$ locally maximizes $\theta \mapsto \mathbb{E}_{x \sim p_{\theta_*}} \log(p_\theta(x))$ and hence locally minimizes $\theta \mapsto D_{\text{KL}}(p_{\theta_*}||p_\theta)$. Hence, $\nabla_\theta D_{\text{KL}}(p_{\theta_*}||p_{\theta_*}) = 0$. Furthermore we know that

$$\nabla_\theta^2 D_{\text{KL}}(p_{\theta_*}||p_\theta) = -\int p_{\theta_*}(x) \nabla_\theta^2 \log(p_\theta) dx$$

For fixed parameters $\theta$, denote for $s \in [0,1]$, $\theta_s = s\theta + (1-s)\theta_*$ and $f(s) = D_{\text{KL}}(p_{\theta_*}||p_{\theta_s})$. We have $f'(0) = 0$ and

$$f''(s) = (\theta - \theta_*)^\top \left( -\int p_{\theta_*}(x) \nabla_\theta^2 \log(p_{\theta_s}) dx \right) (\theta - \theta_*)$$

Using Taylor expansion with explicit remaining, we know the existence of $s \in [0,1]$ such that $f(1) = f(0) + f'(0) + s^2 \frac{f''(s)}{2}$. There remains to bound the spectral norm of $(-\int p_{\theta_*}(x) \nabla_{\theta_s}^2 \log(p_\theta) dx)$. Since by assumption the mapping $\theta \mapsto \mathbb{E}_{p_{\text{data}}} \nabla_\theta^2 \log(p_\theta(x))$ is locally continuous, and that the spectral norm is itself continuous, we know that we can bound on a neighborhood of $\theta_*$, $\|\mathbb{E}_{p_{\text{data}}} \nabla_\theta^2 \log(p_\theta(x))\| \le 2\|\mathbb{E}_{p_{\text{data}}} \nabla_\theta^2 \log(p_{\theta_*}(x))\| := 2C < \infty$. Furthermore, using that $x \mapsto \nabla_\theta^2 \log(p_\theta(x))$ is $L$-Lipschitz (Assumption 2.2) and that $\mathcal{W}(p_{\theta_*}, p_{\text{data}}) \le \varepsilon$ by assumption, using Kantorovitch-Rubinstein duality we know that

$$\left\| \int p_{\theta_*}(x) \nabla_\theta^2 \log(p_{\theta_s}) dx - \int p_{\text{data}}(x) \nabla_\theta^2 \log(p_{\theta_s}) dx \right\| \le \varepsilon L$$

Putting all things together, we know the existence of a constant $C'$ such that for $\theta$ in a neighborhood of $\theta_*$ (that we can in particular choose convex), we have for $s \le 1$, $|f''(s)| \le 2C'\|\theta - \theta_*\|_2^2$ and hence $D_{\text{KL}}(p_{\theta_*}||p_\theta) \le C'\|\theta_t - \theta_*\|^2$ for $C' < \infty$ on a neighborhood of $\theta_*$. Using the previous Lemma A.7, we deduce the convergence rate:

$$D_{\text{KL}}(p_{\theta_*}||p_{\theta_t}) = \tilde{\mathcal{O}} \left( \left( \frac{\lambda(\alpha + \varepsilon L)}{\alpha + \lambda(\alpha - \varepsilon L)} \right)^{2t} \right) .$$

$\square$

## A.5 Experiments

We recall and detail the general set-up of iteratively retraining on a mixture of real data and curated synthetic samples in Algorithm 1

### A.5.1 MoG and two moons datasets - DDPM

**Experimental details**. For both experiments, the learned vector field is parametrized by an MLP of 2 hidden layers and hidden width 128. We use a time discretization in 250 steps. Finally, we retrain the model for multiple iterations (8 for MoG, 5 for two moons), first only on real data and then on *filtered* synthetic samples from the previous iteration using pairwise comparisons. We use

$5 \cdot 10^3$ initial samples from the real data distribution and $5 \cdot 10^3$ generated samples filtered from $10^4$ generated initial samples. When mixing, we use equal fractions of real and filtered samples. For the two moons we add a Gaussian noise with standard deviation $1.10^{-1}$.

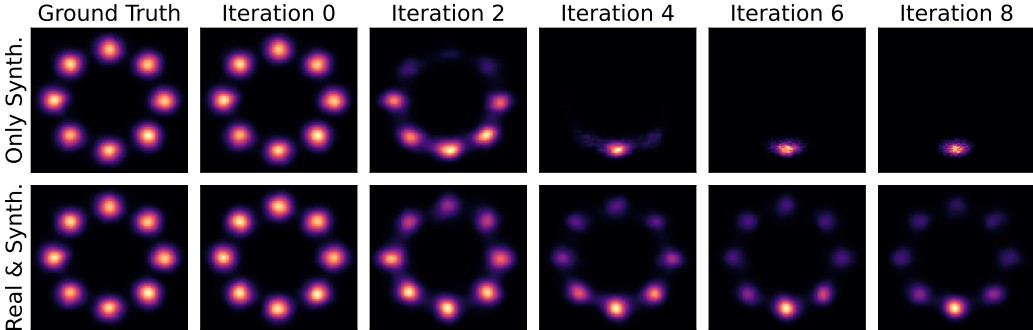

Figure 4: **Mixture of Gaussians.** Iterative retraining on the two moons dataset for 8 iterations. On the top row, we display the fully filtered synthetic loop, and below we use a mixture of real and filtered data.

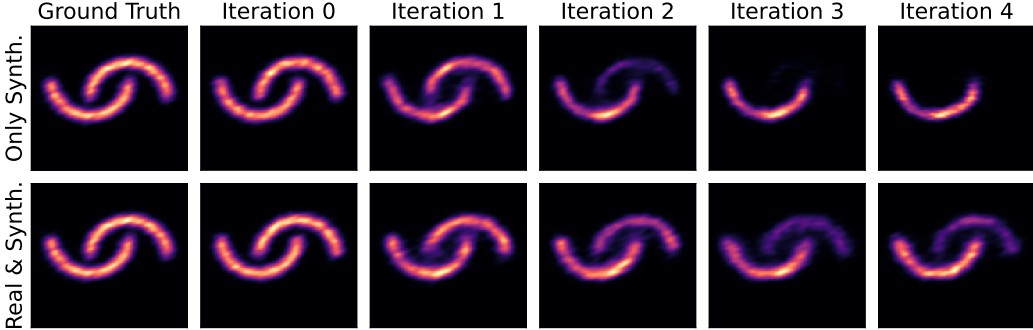

Figure 5: **Two moons.** Iterative retraining on the two moons dataset for 5 iterations. On the top row, we display the fully filtered synthetic loop, and below we use a mixture of real and filtered data.

### A.5.2 Difference between collapse of the reward variance and overall variance

It is crucial to note the difference between collapse of the reward variance and collapse of the overall distribution variance. To highlight this difference, in appendix A.5.2 we show heat-maps of respectively a reward with four different modes, the density of the mixture of Gaussians $p_0$, and the limit density of theorem 2.1 as defined as $p_*(x) := \frac{p_0(x) \mathbb{1}_{r(x)=r_*}}{\mathbb{P}_0(r(x)=r_*)}$. In appendix A.5.2, we show that the reward variance collapses to $0$ while the variance of the overall distribution density does not seem to collapse.

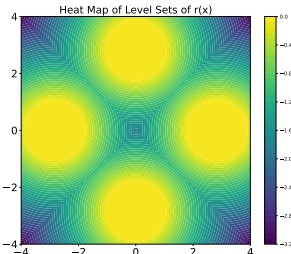

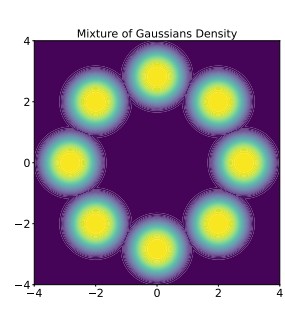

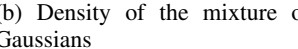

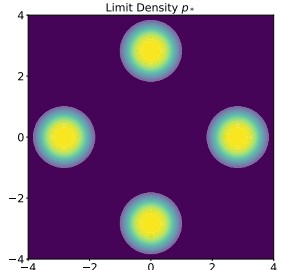

(a) Level sets of the reward function $r(x) = -\max\{0, d(x, \mathcal{D}) - r_{min}\}$ where $r_{min} = 1$ and $d(x, \mathcal{D}) = \min_{y \in \mathcal{D}} \|x - y\|$ and $\mathcal{D}$ is a set of 4 points. In particular, the reward is constant on the 4 balls of radius $r_{min}$ around $\mathcal{D}$

(b) Density of the mixture of Gaussians

(c) Limit density as defined by $p_*(x) := \frac{p_0(x) \mathbb{1}_{r(x)=r_*}}{\mathbb{P}_0(r(x)=r_*)}$ where $p_0$ is the density of the mixture of Gaussians learned at initialization

Figure 6: Plots of respectively a) the level sets of the reward b) the density of the mixture of Gaussians c) The limit density of the fully synthetic retraining loop with curation as predicted by theory

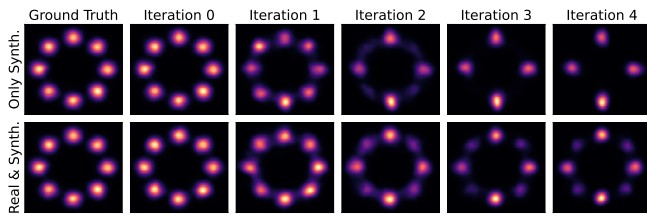

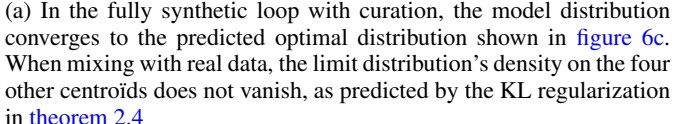

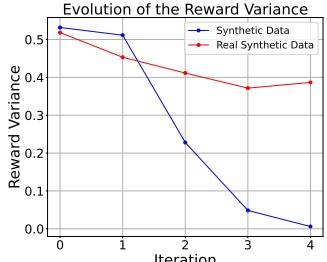

(a) In the fully synthetic loop with curation, the model distribution converges to the predicted optimal distribution shown in figure 6c. When mixing with real data, the limit distribution's density on the four other centroïds does not vanish, as predicted by the KL regularization in theorem 2.4

(b) Evolution of the reward's variance when iteratively retraining the model on either fully synthetic curated data (blue) or on a mix of synthetic and real curated data (red)

Figure 7: Iteratively retraining a diffusion model on a mixture of Gaussians. For curation, we use a reward $r(x) = -\max\{0, d(x, \mathcal{D}) - r_{min}\}$ where $r_{min} = 1$ and $d(x, \mathcal{D}) = \min_{y \in \mathcal{D}} \|x - y\|$ with $\mathcal{D}$ a set of 4 points.

### A.5.3 FID, precision, recall

We measured FID, precision and recall for the three different settings on CIFAR10 presented in Section 4, *i.e.,* a) filtering based on the probability of a classifier on class 0 of planes (Figure 8), b) filtering based on the confidence of the classifier (Figure 9) and c) filtering based on the confidence of the classifier and using a mixture of real data and filtered synthetic samples at each retraining step (Figure 10).

In the first two settings, we observe that the FID dramatically increases during retraining. We want to point out that it is not only due to a degradation in quality of the generated samples but also and mostly from the inequalities of the class proportions emerging during retraining. A clear indicator of this is the correlation between the FID behavior in Figure 8 and the behavior of the proportion of class 0 shown in Figure 2: the FID stabilizes at the end of the retraining loop when the proportion of class 0 reaches its maximum. A second interesting fact is that in all three settings, the precision increases, which hints that filtering does not necessarily degrades the quality of generated samples in our case. Additionally, we can clearly see the impact on stability of real data on Figure 10 where the FID witnesses much smaller variations compared to Figure 9 and Figure 9. Interestingly, we see on Figure 9 that using the confidence of the classifier as a reward function implies a bigger increase

of the precision than on Figure 8 or Figure 10, which correlates with the intuition that confidence is linked to precision. Finally, notice that the three runs on Figure 9 have small variance, as we have already highlighted.

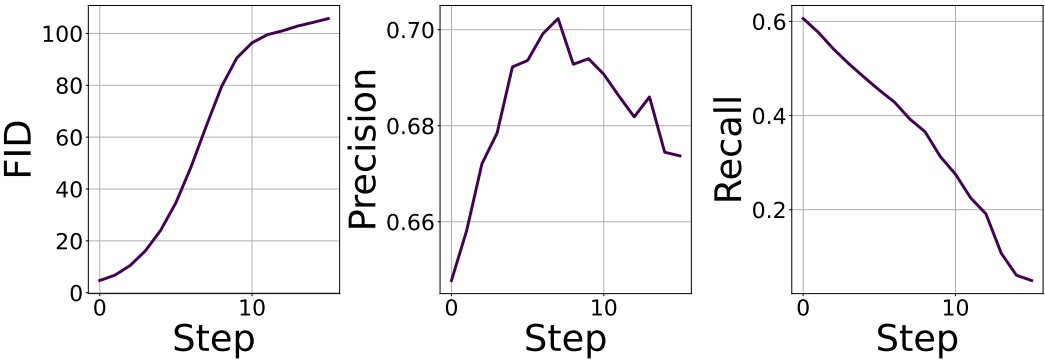

Figure 8: FID, precision and recall when retraining with filtering and $r(x) = -\gamma q_0(x)$, $\gamma = 5$

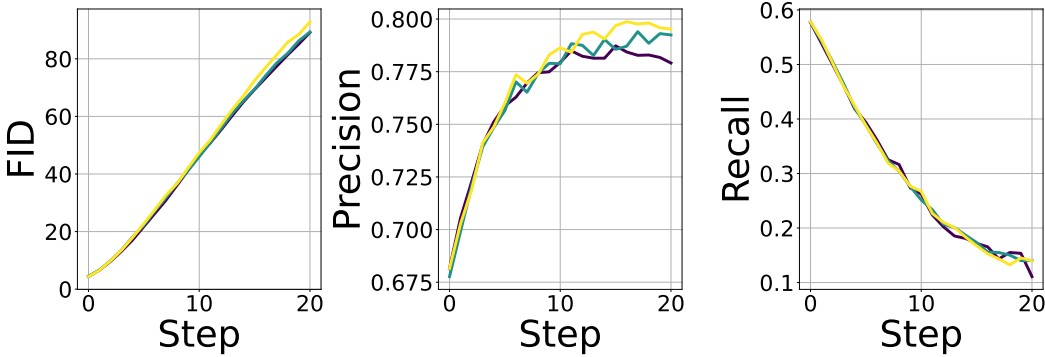

Figure 9: FID, precision and recall when retraining with filtering and $r(x) = \gamma \arg\max_{0 \leq i \leq 9} p_i(x)$, $\gamma = 15$

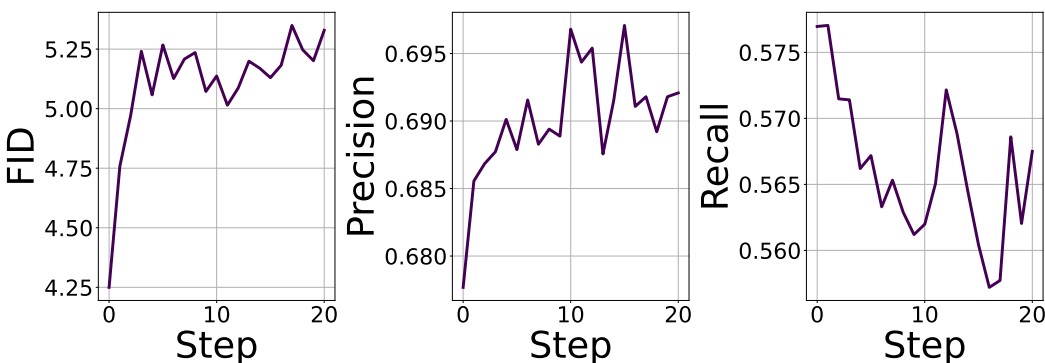

Figure 10: FID, precision and recall when retraining with filtering and $r(x) = \gamma \arg\max_{0 \leq i \leq 9} p_i(x)$, $\gamma = 15$ and reusing real data at each step

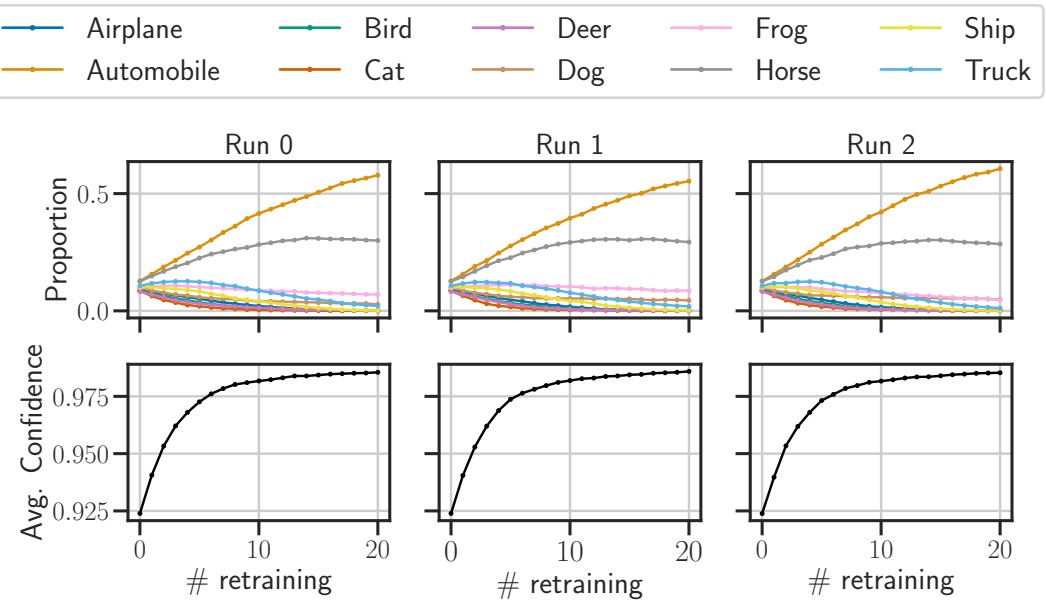

Figure 11: **CIFAR-10**. Evolution of the proportion of the classes and the average reward when filtering based on the confidence of a classifier for three independent runs. The curves have small variance which supports our results when only one run was reported due to the high compute costs of retraining a generative models multiple times.

### A.5.4 Compute Cost

Experiments on synthetic data (mixture of Gaussians and two moons) ran on a single GPU in a few minutes. However, retraining with filtering on CIFAR10 was more costly. On a A100 GPU of 40GB RAM and using 4 workers with total 32 GB RAM, retraining for 20 iterations with generation of 50000 samples took about 22 hours.

### A.6 Examples of sets of four generated images on MidJourney and Stable Diffusion 2.1

We show in Figure 12 two sets of four images generated respectively with Midjourney and Stable Diffusion. In both cases, users can choose their preferred image out of the 4 proposed and more specifically in the case of Midjourney, *upscale* it.

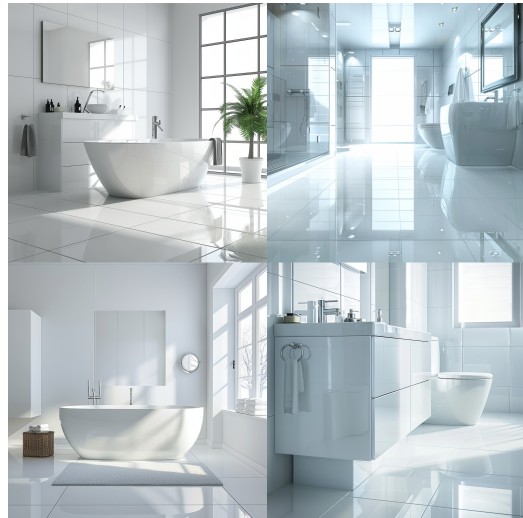 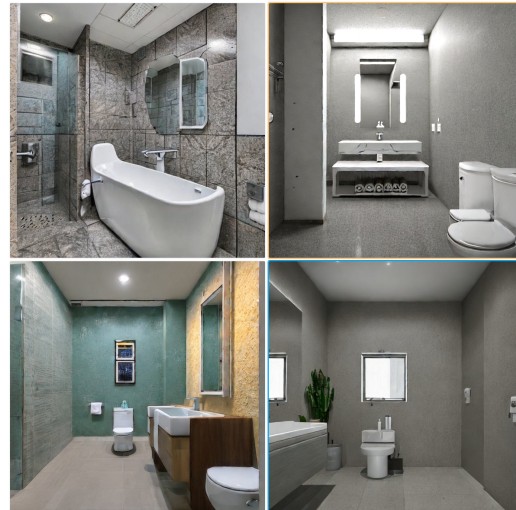

(a) **Midjourney.** Images from Midjourney discord, generated with the prompt "Modern and white bathroom, clean and shiny, high resolution, a real scene".

(b) **Stable Diffusion.** Four images were generated using Stable Diffusion 2.1 Hugging Face implementation (Hug), with the prompt "a bathroom'".

Figure 12: Two sets of four images were generated using two different generative models. For Midjourney (Figure 12a), users can select which image to *upscale*. The upscaled images are then incorporated into the JourneyDB dataset (Pan et al., 2023). For Stable Diffusion, users can choose the preferred generated image.

