# OpenReview forum: "Self-Consuming Generative Models with Curated Data Provably Optimize Human Preferences"
_NeurIPS.cc/2024/Conference — NeurIPS 2024 spotlight_

### Official Review · Reviewer_orKH · 2024-07-08

**Soundness:** 3
**Presentation:** 2
**Contribution:** 4
**Rating:** 7
**Confidence:** 4

**Summary:**

The authors investigate the properties of self-consuming loops that arise in the training of generative models. In particular, they investigate the impact that data curation has on the iterative retraining performance of these models. The paper contains theoretical and empirical analysis of how model performance is affected by various data curation assumptions (only synthetic examples, or real data being injected at each step, or human-preference curated synthetic samples, etc).

**Strengths:**

Originality: I believe this work is highly original, largely because it considers a new problem formulation--"what happens to self-consuming generative models when the synthetic data that they re-train one has been curated via human preferences?". To my knowledge, nobody has considered this problem, and it is an excellent and timely problem to consider.

Clarity: The authors motivated the problem very clearly.

Significance: many previous works have investigated self-consuming loops, but the area is still burgeoning, and right now, the area seems like "the wild west"--there are lots of papers out there with different assumptions, different results, no agreed-upon benchmarks tasks, etc. This paper is significant because it considers a more realistic setup than previous papers; it considers the setting where the web-scale data contamination happens because of human-curated data. This is an important case to consider, and represents a large step towards modeling the data contamination issue more rigorously. This more realistic setup comes with more mathematical overhead, which is challenging to deal with.

**Weaknesses:**

Clarity: in my opinion, the presentation of statements of the theorems in the paper needs improvement in order to be useful to the community. Consider for example Theorem 2.1. The assumptions for this theorem are distributed in the section preceding it, which makes it much harder to understand and contextualize that theorem. I would strongly suggest summarizing the assumptions and key notations in the statement of that theorem (and likewise for all the other results.) An excellent model for this would be the paper that the authors cited the most, Bertrand et al's "On the Stability of ..." . That paper's Theorem 1 (from the latest arXiv version) begins--"Given theta^\star as defined in Equation (7) that follows assumptions 1 and 2....". Given that a large part of this paper's contribution is its theorems, I would be likely to raise my score, and champion this paper, if I could first see an updated draft which makes explicit every assumption in the statement of each theorem.

See also the limitations section.

**Questions:**

I'm not sure I understand how the experiment supports the theory here, please help clarify this for me--at line 250, the authors say "applying theorem 2.1, the density will converge to a renormalized Gaussian distribution restricted to the ball centered at x_* of radius r_min". But this isn't what I see in Figure 4, it looks like there are a bunch of different Gaussian balls with different densities, corresponding to different intensities. Should these gaussian balls all be the same density, or did I misunderstand something?

Some misc points that didn't affect my score/judgment of the paper, but the authors should consider fixing in the next draft:
- improper formatting in equation 8 (spacing/parentheses)
- awkward/grammatically incorrect wording in the sentence immediately after equation (7)

**Limitations:**

The two main things preventing me from giving a higher score are the following two limitations. I'm looking forward to hearing the author's rebuttal to these points (and hopefully seeing an updated draft, if possible).

1. Presentation of statements of theorems (what are the specific hypotheses? eg take a look at the latest version of the bertrand et al paper on the arXiv for a good way to do this, since that paper is structured in a similar way. They number their assumptions and make them more clear. See "Weaknesses" section for more details.) This is the primary limitation, from my perspective, as I think the usefulness of the paper is very limited by unclear theorem statements.

2. Proper contextualization of these results relative to the literature--namely, there's a difference between the work in Bertrand et al and Alemohammad et al, but the authors seems to be comparing them in an "apples to apples" way. Namely, the former work considers the case of iterative fine-tuning, whereas the latter considers the case of retraining from scratch. In the former case, it is strictly easier to avoid model collapse, since the model parameter update is "local", and in the latter case, the updates are "global." Specifically, on line 218 it says: "Alemohammad et al. (2024); Shumailov et al. (2023) first evidenced catastrophic degradation of the generated data in the fully synthetic loop. Bertrand et al. (2024) mitigate these conclusions in the setting where the model is retrained on a mixture of synthetic and real data and they show the stability of the process around the data distribution." I think this is a false statement, because Alemohammad et al considered re-training from scratch at each iteration, which wasn't considered in the Bertrand et al paper--but please correct me if I'm misunderstanding. And in that same vein, I think it is important to properly contextualize the present paper in the presence of that dichotomy--does this paper consider iterative fine-tuning, or iterative re-training from scratch? That is important information for the readers/the literature.

---

> ### Author Rebuttal · Authors · 2024-08-06
>
> We thank the reviewer for their constructive comments and are pleased that they find the question raised "highly original" and "very clearly" motivated. We further appreciate that they find it "significant" and "more realistic" than previous works. We now address the key clarification points raised by the reviewer.
>
> ## Presentation of statements of the theorems
> We acknowledge the reviewer's concern that the dissemination of all of our assumptions may not have been sufficiently clear. Below, we present updated assumptions and theorems that we hope address the clarity issue.
>
> We propose to restructure assumption 2.1 to sub-assumptions of increasing strength:
>
> **Assump. 2.1-A:** The distribution $p \in \mathcal{P}(\mathbb{R}^d)$ has a density w.r.t. Lebesgue measure and $E_p[e^{r(x)}] < \infty$.
>
> **Assump. 2.1-B:** The distribution $p\in \mathcal{P}(\mathbb{R}^d)$ has a density w.r.t. Lebesgue measure and there exists $r_* \in \mathbb{R}$ such that: (a) $p$-almost surely, $r(x)\leq r_*$ and (b) $p$ puts positive mass in a neighborhood of $r_*$ i.e., $\forall \epsilon > 0, P_0(r(x)\geq r_* - \epsilon) > 0$.
>
> **Assump. 2.1-C:** Assum. 2.1.B and $P(r(x) = r^*) > 0$.
>
> Using these assumptions, our results of section $2$ will have the following overheads:
>
> > **Thm. 2.1:** Let for all $t\geq 0$, $p_{t+1}$ be the distribution induced from a discrete choice model on $p_t$ (4) where $\mathcal{P}= \mathcal{P}(\mathbb{R}^d)$ is the set of probability distributions on $\mathbb{R}^d$. If $p_0$ satisfies Assump 2.1-C, then we can define $p_*(x):=\frac{p_0(x)1_{r(x)=r_*}}{P_0(r(x)=r_*)}$ and the self-consuming loop on curated samples $p_t$ converges to $p_*$: $KL(p_*||p_t) \xrightarrow{t\rightarrow \infty} 0$
>
> The other results have been updated similarly. If the reviewer would like to see the other exact statements, we can share them in another response.
>
> ## Contextualization of our results in the literature on model collapse
> We agree with the reviewer that better contextualization of our results on whether the retraining is performed from scratch or via fine-tuning can improve the clarity. **We will update the paper to clarify this and present below how we would proceed.**
>
> ## On retraining from scratch vs iterative fine-tuning:
>
> **1.) Experiments:** All retraining step in experiments on mixture of Gaussians (MoG) and two moons are performed from scratch, whereas in the case of CIFAR dataset, due to the high compute cost of retraining the model from scratch (20 hours on an A100 GPU) we performed fine-tuning at each step. Fine-tuning is always performed on $10^6$ images. We use the same amount of images for fair comparison between different proportions of real data injected. In contrast, in [1], the collapse is shown when the model is retrained from scratch at each iteration. In [3], the experiments are performed using retraining from scratch for VAEs and GMMs and sequential fine-tuning for LLMs. In [2], toy experiments on two moons and MoG are performed by retraining from scratch, while experiments on CIFAR10 and FFHQ are performed using iterative fine-tuning. We agree with the reviewer's remark that stability using real data in the setting of iterative fine-tuning is easier to obtain than when retraining from scratch since the model parameters are initialized around a good potential set of parameters. However, we would also like to point out that model collapse occurs also in the setting of iterative fine-tuning as shown in Figure 2 of [2] (red curves).
>
> **2.)Theory:** Finally regarding our theoretical results, only thm 2.2 happens in the setting of iterative fine-tuning (since it uses the same setting as in [2]). However, all our other results and in particular thm 2.1, 2.3, 2.4 do not explicitly assume a special learning algorithm in parameter space. Instead, we consider having a perfect learning model and consider the evolution of the expected reward for such a learning model. In that sense, it applies to learning from scratch with the additional assumption that the model attained perfectly fits the curated distribution. We will make this point clear in the updated draft.
>
> ## On fresh real vs fixed real data:
> [1] studies the self-consuming loop in three different settings where the model is retrained a) only on synthetic data b) on a mixture of synthetic data and a fixed set of real data samples b) on synthetic data and a fresh set of real data samples at each step. In setting (a), the retraining loop collapses. In (b), it collapses, too, but with some delay related to the amount of fixed real data. In (c) the retraining loop does not degrade performances provided there is enough fresh real data at each step. [2] proved stability in setting (b) under some theoretical assumptions and in the iterative finetuning framework. Comparatively, our experiments on MoG are performed using fresh real data at each step while the CIFAR experiments are performed in the fixed real data framework.
>
> ## Experiment clarification:
> We thank the reviewer for highlighting a relevant point on the presence of multiple spots with different densities of fig. 4 at iteration 4. These are remaining errors due to the non-fully convergence. To improve clarity, we ran the experiment for more iterations (global response fig 3) in which case it is clearer that the final distribution is a unique renormalized Gaussian distribution restricted to the ball centered at $x_*$ of radius $r_{min}$.
>
> If the reviewer finds our updates agreeable, we would appreciate it if the reviewer considers championing/upgrading their score for this paper, as mentioned in the original review. We are also happy to answer any further questions that the reviewer may have.
>
> [1] Alemohammad et al Self-consuming generative models go mad. ICLR 2024
>
> [2] Bertrand et. al On the Stability of Iterative Retraining of Generative Models on their own Data. ICLR 2024
>
> [3] Shumailov et al. The curse of recursion: Training on generated data makes models forget. Nature 2024

---

> ### Comment · Reviewer_orKH · 2024-08-12
> **Response to rebuttal**
>
> I would like to thank the authors for the thoughtful reply to my review.
>
> In particular, I appreciate the additional contextualization of works [1] and [2]; I think that the literature would greatly benefit from having that explanation from the section "On fresh real vs fixed real data". I also appreciate the more clear statement of Theorem 2.1.
>
> Would it be possible to share here the other exact statements as well--namely, how the authors would propose to update the other exact statements of the other theorems, 2.2, 2.3, 2.4--in another response?
>
> **I have updated my score to above the acceptance threshold, with the expectation that the authors will update the camera-ready version's main result statements with more clearly stated hypotheses and clearly referenced terms, similar to the rebuttal above.** Although my concern was mainly with Theorem 2.1 (it contained the least context out of all the other theorems), in my opinion, Theorems 2.2, 2.3, 2.4 should each be improved via clearer context, and I would want to see the updated statements if possible.
>
> I strongly believe that having these statements stated that clearly would better allow the community to benefit from the authors' work.

---

> > ### Author Response · Authors · 2024-08-13
> > **Response to the Reviewer's Comment**
> >
> > We are grateful to the reviewer for their valuable feedback and for increasing their score. We will include in the updated manuscript the new presentation of the theorem statements with their assumptions, along with the contextualization of our results in the literature. We present below how we aim to update the statements of Assumption 2.2 and theorems 2.2, 2.3, and 2.4.
> >
> > **Assumption 2.2:** For $\theta$ close enough to $\theta_*$, the mapping $x \mapsto \nabla^2_\theta\log p_\theta(x)$ is $L$-Lipschitz and the mapping $\theta \mapsto E_{p_{data}}\left[\log p_\theta(x) \right]$ is continuously twice differentiable with $E_{p_{data}}\left[\nabla^2_\theta\log p_\theta(x)\right] \preceq -\alpha I \prec 0$. Further suppose $W_1 (p_{\theta_*}, p_{data})\leq \epsilon$, i.e. $p_{\theta_*}$ is close to the data distribution $p_{\text{data}}$.
> >
> > **Theorem 2.2:** Under Assumption 2.2, if $L \epsilon < \alpha$ and $\lambda<\frac{\alpha}{2L\epsilon}$, then there exists a neighborhood of the optimal distribution parameters $\theta_*$ such that for any initial parameters $\theta_0$ in that neighborhood, $p_{\theta_t}$ converges to $p_{\theta_*}$ exponentially fast:
> >
> > $$ KL(p_{\theta_*}||p_{\theta_t}) = \tilde{\mathcal{O}}\left(\left(\frac{\lambda(\alpha+\epsilon L)}{\alpha+\lambda(\alpha-\epsilon L)}\right)^{2t}\right)$$
> >
> >
> > **Theorem 2.3:** Let $\lambda > 0$ and consider the process $(p_{t})$ defined in eq. 8, with $p_0 = p_{ref}$. If $p_{ref}$ satisfies Assumption 2.1 B, then for all $t\geq1$:
> >
> > $$E_{p_t}\left[e^{r(x)}\right] \geq E_{p_{ref}}\left[e^{r(x)}\right] + \frac{\lambda}{(1+\lambda)^3}\frac{(K-1)Var_{p_{ref}}\left[e^{r(x)}\right]}{Ke^{r_*}}$$
> >
> > **Theorem 2.4:** Let $\lambda > 0$ and $p_{ref}\in \mathcal{P}(\mathbb{R}^d)$ with a density w.r.t. Lebesgue measure. Consider the process $(p_{t})$ defined in Equation 8, with $p_0 = p_{ref}$. Suppose that $\lambda < \frac{1}{K-1}$, then, for all $t\geq1$:
> >
> > $$KL(p_t||p_{ref})\leq -\log\left({1-\lambda(K-1)}\right)$$
> >
> > We hope this answers the reviewer’s remaining concerns and are happy to provide clarification to any additional questions the reviewer may have.

---

### Official Review · Reviewer_cgiY · 2024-07-10

**Soundness:** 3
**Presentation:** 2
**Contribution:** 2
**Rating:** 6
**Confidence:** 4

**Summary:**

This paper studies the impact of data curation on iterated retraining of generative models. Theoretical results are derived for the convergence state of the retraining loop when using a fraction of curated synthetic data or a mixture of real data and curated synthetic data at each step. Empirical experiments on both synthetic datasets and CIFAR-10 demonstrate that the proposed approach can bias the generative model to generate samples with higher reward.

**Strengths:**

The problem of iterated retraining of generative models using curated data is important and interesting. The authors have indeed made some achievements in this direction. The theoretical results presented in this paper are interesting and reasonable, especially in their connection to preference optimization. The toy experiments are consistent with the theoretical claims.

**Weaknesses:**

The writing of this paper needs further improvement. Some notations are unsuitable, and certain mathematical notations are introduced without explanation. Some literature references are missing.
1. In Eq.2, $\mathcal{BT}\left(x_1, \ldots, x_K\right)$, where "BT" refers to the Bradley-Terry model, is typically used to model pairwise preferences, whereas the Plackett-Luce (PL) model is proposed for preferences involving more than two items.
Eq.5: If $p_{t+1}(x)$ is not a normalized density function, the same applies to $p_{t+1}(x)$ in Eq.6.
2. There are already studies on deep generative models (GANs) that investigate the convergence state of iterative retraining on curated data from the perspective of preference [1, 2]. It should include a discussion of these works.
3. The theoretical proofs especially the part in Section 2.2 are heavily inherited from the previous work [3], which somehow makes the theoretical contributions marginal.

[1] Gupta, A., Zou, J. Feedback GAN for DNA optimizes protein functions. Nat Mach Intell. 2019.
[2] Yao, Y., Pan, Y., et. al. Differential-Critic GAN: Generating What You Want by a Cue of Preferences. IEEE Transactions on Neural Networks and Learning Systems, 2022.
[3] Bertrand, Q., Bose, et. al. On the Stability of Iterative Retraining of Generative Models on their own Data. In The Twelfth International Conference on Learning Representations.

**Questions:**

1. Eq.3: The term $p_{ref}$ is not explained. What distinguishes $p_{ref}$ from $p_{data}$?
2. In Theorem 2.2, deriving $\lambda < 0.5$ based on the assumption $L \varepsilon < \alpha$ and $\lambda < \frac{\infty}{2 D \varepsilon}$ appears to conflict with the claim in line 181.
3. The explanation of $r_{\ast}$ in lines 134-137 is somewhat complex. Adding some equations would clarify its meaning.

**Limitations:**

Yes.

---

> ### Author Rebuttal · Authors · 2024-08-06
>
> We thank the reviewer for their time and constructive comments. We appreciate that the reviewer finds the problem we tackle "important and interesting" and that our theory in connection to preference optimization “interesting and reasonable”. We now address the key points raised in the review:
>
> ## Plackett-Luce model
> We thank the reviewer for referring us to the Plackett Luce model. We would like to mention that we were aware of the Luce choice rule model that we cited on lines 87-90. We will change our notation from $BT$ to $PL$ referring to the Plackett-Luce model since it provides a more unified framework consistent with the literature.
>
> ## Normalization of $p_t$
> We agree with the reviewer on the fact that **if** $p_t$ is not normalized, then $p_{t+1}$ is not normalized either. However we do not foresee a problem since in practice we apply our results to an **initial normalized probability distribution**. We hope this answers the reviewer’s question and are happy to clarify further.
>
> ## Related Work
> We are grateful to the reviewer for referring us to the two important references [1, 2] that we will add to our related work section. In [1], the authors tackle the problem of generating synthetic DNA sequences using GANs. They introduce an external function analyzer to rate synthetic samples from the generator and add the highest-scored ones into the discriminator training set. *This work [1] is mostly experimental and tied to the GAN architecture while we adopt a more theoretical framework to understand the self-consuming loop without specifying any architecture*.
>
> Furthermore our study takes the point of view of the recently developed model collapse literature and relies only on $K$-wise preferences as in [2]. In [2], the authors propose a new GAN framework to incorporate user’s preferences in the training. They show state of the art results in generating the user-desired data distribution and theoretically prove the convergence of their method. *The key difference to our work is that they aim to generate a diversity of samples that are desired by users and their focus is therefore not on the collapse of their method to a maximal reward set.*
>
> ## Closeness with [3]
> We acknowledge the reviewer’s comment that our proof of theorem 2.2 is adapted from the proof of [3], since our goal was to improve their paper’s main result in the same setting. Note that despite the similarity in the proof technique **the theoretical improvement we provide is significant**.
>
> Moreover, we believe that it does **not** consist in our main theoretical contributions, and was serving as an introduction to the rest of the section on how real data provides stability to the retraining loop. Instead, our major theoretical results, i.e. **theorem 2.1, 2.3 and 2.4 are proved in a different setting** and with proof techniques that completely differ from [3].
>
> ## Clarification of the $p_{ref}$ notation
> We acknowledge the reviewer's concern about the notation $p_{ref}$ and we will clarify this notation in the updated draft. Namely, we denote $p_{ref}$ any probability distribution with density with respect to Lebesgue measure. It may indeed be of interest to apply thm 2.3 to other cases than retraining on a mixture of synthetic data at iteration $t$ and the data distribution. For example, it may be of interest to retrain on a mixture of synthetic data at iteration $t$ and of the synthetic data at initialization. In such a case, $p_{ref}$ would be identified to $p_0$. It bridges the gap with RLHF, as the KL regularization in the RLHF objective is done with respect to the supervised fine-tuned policy (see equation 3 in the DPO paper [5]). This makes our result more general as we show that stability occurs around any reference distribution that is injected in the retraining loop (either real data samples or samples from the initial model). It also provides an avenue to study the retraining process over a mixture of all previous iterations of the generative models.
>
> ## Clarification on the assumption on $\lambda$
> We respectfully disagree with the reviewer that our assumptions $$L\epsilon < \alpha \quad \text{and} \quad \lambda < \frac{\alpha}{2L\epsilon}$$ are in conflict with the claim that previous work was restricted to $\lambda < \frac{1}{2}$. Indeed, the necessary conditions for theorem 1 of [3] are $$\lambda \leq \frac{1+ \frac{L\epsilon}{\alpha}}{2}\quad \text{and}\quad L\epsilon< \alpha$$  which necessarily requires $\lambda < \frac{1}{2}$. These conditions are more restrictive than ours. In particular, we see that for fixed $\alpha$, if $L\epsilon\rightarrow 0$ then both our conditions are satisfied (in particular $\lambda$ bigger than $\frac{1}{2}$ will be allowed).
>
> ## Clarification of $r_*$ notation
> We agree upon the reviewer's remark that our definition of $r_*$ could be clarified and we will improve this paragraph in the updated paper. In a nutshell, $r_*$ should be thought as the smallest number that upper-bounds the random variable $p_0$ with probability $1$. For example, a Uniform distribution on the interval $[0, 10]$, $r_*=10$ whereas for unbounded distributions such as $\mathcal{N}(0, 1)$, $r_*$ does not exist. As suggested by the reviewer, an equation that could clarify this is $r_* = inf \\{r \in \mathbb{R}, \mathbb{P}(r(x)\leq r_*) = 1\\}$.
>
>
> We thank the reviewer for their valuable feedback and great questions. We believe we have answered to the best of our ability all the great questions raised by the reviewer and we kindly ask the reviewer to potentially upgrade their score if they are satisfied with our responses. We are also more than happy to answer any further questions that arise.
>
> [4] Gerstgrasser, Matthias, et al. "Is model collapse inevitable? breaking the curse of recursion by accumulating real and synthetic data." arXiv 2024
> [5] Rafailov, Rafael, et al. "Direct preference optimization: Your language model is secretly a reward model." NeurIPS 2023

---

> > ### Comment · Reviewer_cgiY · 2024-08-11
> > **Thanks for the responses.**
> >
> > I appreciate the authors' clarification. I increased my score to "weak accept".
> >
> > About $p_t$. Usually, the probability is normalized to 1. If not, it should add the clarification.

---

### Official Review · Reviewer_ej3h · 2024-07-12

**Soundness:** 3
**Presentation:** 3
**Contribution:** 3
**Rating:** 6
**Confidence:** 4

**Summary:**

This paper extends Bertrand et al. 2024’s analysis of model collapse to study settings where data is filtered (based on a particular preference model) before being used for training the next iteration of generative models.

**Strengths:**

- The paper extends prior work studying self-consuming generative models to integrate preference learning, which is highly sensible
- The paper is well written

**Weaknesses:**

- I do not think that Equation (3) faithfully models reality. Specifically, it assumes that the $t+1$-th model is fit to a mixture of (1) real data and (2) preference-filtered data from the $t$-th model. But realistically, synthetic data should amass over time, as I believe is the case with the datasets that the authors mention in their introduction, e.g., LAION-5B. This is a point made by "Is Model Collapse Inevitable? Breaking the Curse of Recursion by Accumulating Real and Synthetic Data." and I agree with them.

- I might be missing something, but intuitively, Theorem 2.1 seems wrong, specifically the claim “eventually converges to the highest level set of the reward reached **at initialization**”. For the simplest possible counter example, suppose p_0 is a discrete Uniform distribution on integers 0 to 10 and the reward function is $r(x) = x$. Defining $r_* := 10$, we see that Assumption 2.1 is satisfied, including the variant where $\epsilon \geq 0$. It is hard to imagine that the iterative loop would concentrate on $10$ rather than something greater than $10$. The intuition is that sampling and filtering from the first model iteration might shift the distribution towards regions where it is possible to sample higher rewards.

- I think the experiments corresponding to Theorem 2.1(Figures 4 and 5) could be improved to better connect with the maths. Specifically: (1) Provide a heatmap of the level set of $r_*$ to show what $p_*$ is. (2) As I understand, the theorem doesn’t say that the distribution’s variance collapses, but rather, that the _reward variance_ collapses. In the figures, the reward functions are unimodal, and so we see the distributions converge towards unimodal behavior. You should modify the reward functions to be multimodal to demonstrate this distinction between the distribution's variance collapsing versus the reward's variance collapsing. One way to do this might be to define the reward function as 4 of the 8 MoGs (i.e. the reward is the max of the set of negative distances to the 4 chosen centroids). Then, we should see the model concentrate on those 4 chosen centroids.

- I feel like I don’t understand Theorems 2.3 or 2.4. In Theorem 2.3, the lower bound on the right hand side does not appear to depend on the model fitting iteration $t$, as best as I could tell, nor does the upper bound in Theorem 2.4. Here, I’m expecting the answer to depend heavily on the model fitting iteration. I read the adjacent discussion but didn't receive any clarity. I'm consequently not sure how to evaluate the significance of Section 2.2. Perhaps the authors could clarify?

- I think there are many additional papers you might want to cite. Some may be concurrent with yours (I intentionally did not search for a preprint in order to preserve double blind reviewing), and if so, that’s fine. Here are some suggestions on several different topics:

**On Model Collapse:**

Beyond Model Collapse: Scaling Up with Synthesized Data Requires Reinforcement. https://arxiv.org/abs/2406.07515

Is Model Collapse Inevitable? Breaking the Curse of Recursion by Accumulating Real and Synthetic Data.
https://arxiv.org/abs/2404.01413

**On mode collapse in RLHF:**

Understanding the Effects of RLHF on LLM Generalisation and Diversity. https://openreview.net/forum?id=PXD3FAVHJT

A Distributional Approach to Controlled Text Generation.
https://openreview.net/forum?id=jWkw45-9AbL

Red Teaming Language Models with Language Models
https://arxiv.org/abs/2202.03286

Improving alignment of dialogue agents via targeted human judgements
https://arxiv.org/abs/2209.14375

Aligning Language Models with Preferences through f-divergence Minimization
https://arxiv.org/abs/2302.08215

**On filtering data using reward models - often known by multiple names in the RLHF literature including “Best of N” or “rejection sampling” or “reranking” in the RLHF literature:**

Scaling Laws for Reward Model Overoptimization
https://arxiv.org/abs/2210.10760

Training a Helpful and Harmless Assistant with Reinforcement Learning from Human Feedback
https://arxiv.org/abs/2204.05862

There are many more

**Questions:**

N/A

---

> ### Author Rebuttal · Authors · 2024-08-06
>
> We thank the reviewer for their nuanced review and constructive feedback. We appreciate that the reviewer has found our paper "well-written" and the integration of preference learning to the model collapse literature to be "highly sensible". We took note of the reviewer's concerns and provide clarifications:
>
> ## Accounting for the accumulation of data
> We thank the reviewer for referring us to the paper [1] and pointing out this interesting extension. We believe, together with the reviewer, that the setting of [1] could be adapted to show that accumulating data provides additional stability to the retraining loop and avoids collapse. However, we believe such a study is out of the scope of our work whose aim was to introduce and theoretically develop a new research question of model collapse from the view of preferences. We will, therefore, update our draft to provide additional clarification of this aspect and mention it as an exciting future direction.
>
> ## Clarification on Thm 2.1
> We understand the reviewer's natural concern. The crucial point is that the curated distribution at time $t+1$, $p_{t+1}$ is issued from learning a modified distribution from $p_t$ constructed by **sampling from** $p_t$ and curating samples using preferences. This implies that if a set has probability $0$ for $p_t$, it will never be sampled and therefore never be preferred over other samples. This means that this set will also have probability $0$ for $p_{t+1}$.
>
> Finally, we note that the reviewer’s intuition is valid when the retraining step is not perfect (for example due to bounded expressivity of the class $\mathcal{P}$ involved in equation 4), or when noise is injected in the process. Then the support of $p_{t+1}$ is not necessarily included in the support of $p_t$ anymore. In that case, the self-consuming loop iteratively explores regions that had probability $0$ at initialization and converges to the maximal reward possible, validating the reviewer’s intuition.
>
> ## Improvement of the experiments on MoGs
> We thank the reviewer for their suggestion which will help clarify the theoretical results. We provided in the additional pdf for figures a heat map of the level set of the reward when using $4$ centroïds (Fig 1a). We additionally provided a heat map of the raw mixture of Gaussians (MoG) distribution (Fig 1b) and the corresponding limit distribution defined L154, as the renormalized MoG restricted to the set of maximal reward at initialization (Fig 1c).  We also re-ran our experiments for this reward model and observed that the learned distribution converges as expected to $4$ Gaussian restricted to $4$ balls around the designed centroïds (Fig 2a). We additionally plotted the reward variance and showed that it vanishes in the purely synthetic setting. This demonstrates that the reward variance can vanish independently of the overall distribution variance. We will clarify this in the updated manuscript as it is central to our analysis.
>
> ## Clarification of Thm 2.3, 2.4
> The goal of these two theorems is to respectively provide a lower bound on the expected reward and an upper bound on the KL divergence of a self-consuming generative model loop **with respect to a reference distribution $p_{ref}$**. These theorems formally connect our results to the KL regularization in RHLF ensuring that the KL divergence between the aligned model and the reference model is not too large. This is why only $p_{ref}$ and not $p_t$ appear on the right-hand side. The reviewer’s observation is, however, insightful as our proof proceeds by an induction argument in which each induction step uses Lemma A.1 (see line 563) and hence a right-hand side dependent on $t$. We finally achieve to make this term not dependent on $t$ by induction.
>
> ## Related work
> We thank the reviewer for pointing us to these important references that we will cite and discuss in our revision. We will especially provide more discussion on the relationship between RLHF and rejection sampling fine-tuning. We now discuss the most relevant works mentioned by the reviewer (because of the space constraint we do not have space to discuss them all extensively here):
> Note that [2,3] are concurrent work according to NeurIPS guidelines:
> >papers that appeared online within two months of a submission will generally be considered "contemporaneous"
> The key difference between [2] and our work is that they show the benefit of using feedback on the quality of a sample to prevent model collapse.
>
> In [3], the authors investigate how the different stages of alignment affect a model’s generalization capabilities and output diversity. They empirically show that the output diversity of the RLHF policy is decreased w.r.t. the supervised finetuned policy, which is consistent with our theoretical insights (e.g. lem. 2.2, thm 2.1). However, there are major differences with our setting as their contribution is empirical, and we investigate the impact of iteratively retraining a model several times on synthetic samples while they study a single training round using RLHF.
> [4] experimentally investigates the self-consuming loop, specifically in the case of LLMs, and evidences model collapse in that setting.
>
> We thank the reviewer again for their review and detailed comments that helped strengthen the paper. We hope our answer here and in the global response allows the reviewer to consider potentially upgrading their score if they see fit. We are also more than happy to answer any further questions.
>
> [1] Gerstgrasser, Matthias, et al "Is model collapse inevitable? breaking the curse of recursion by accumulating real and synthetic data" 2024
>
> [2] Feng, Yunzhen, et al "Beyond Model Collapse: Scaling Up with Synthesized Data Requires Reinforcement" 2024
>
> [3] Kirk, Robert, et al "Understanding the effects of RLHF on LLM generalisation and diversity" 2023
>
> [4] Briesch, Martin, et al "Large language models suffer from their own output: An analysis of the self-consuming training loop" 2023

---

> ### Comment · Reviewer_ej3h · 2024-08-12
> **Response to Authors' Rebuttal [Part 1]**
>
> Thank you to the authors for their response!
>
> > Improvement of the experiments on MoGs
>
> > We provided in the additional pdf for figures a heat map of the level set of the reward when using centroïds (Fig 1a).
>
> This is wonderful. Thank you for running these additional experiments - I really like them, and I think they'll improve the paper (in my opinion; if you disagree, you don't need to include them).
>
> > Note that [2,3] are concurrent work according to NeurIPS guidelines:
>
> Yes, then it's fine to not cite them. I tried to allude to that above ("Some may be concurrent with yours") but apparently didn't finish the sentence. I appreciate your care to the other citations.

---

> ### Comment · Reviewer_ej3h · 2024-08-12
> **Response to Authors' Rebuttal [Part 2]**
>
> > We understand the reviewer's natural concern. The crucial point is that the curated distribution at time $t+1$, $p_{t+1}$ is issued from learning a modified distribution from $p_t$ constructed by sampling from $p_t$ and curating samples using preferences. This implies that if a set has probability $0$ for $p_t$, it will never be sampled and therefore never be preferred over other samples. This means that this set will also have probability $0$ for $p_{t+1}$.
>
> I'm not sure I buy this argument. I agree that if a set has probability $0$ for $p_t$, it will never be sampled and therefore will never be preferred over other samples. But you lose me for two reasons:
>
> 1. Depending on the choice of realizable distributions $\mathcal{P}$, the support might be the entire space of outcomes e.g., if $\mathcal{P}$ is the set of Gaussian distributions. In this case, no set would have probability $0$ for $p_t$. Your explanation then seems to hinge on a condition "if" that might not be applicable.
>
> 2. Even if the conditional statement is true i.e. there is some set with mass/density 0 under $p_t$, why is $p_{t+1}$ prohibited from extending its support to this set? In general, probabilistic models are often capable of placing mass/density on sets that were not in their training data.
>
> Could the authors please clarify?

---

> ### Comment · Reviewer_ej3h · 2024-08-12
> **Response to Authors' Rebuttal [Part 3]**
>
> > We thank the reviewer for referring us to the paper [1] and pointing out this interesting extension. We believe, together with the reviewer, that the setting of [1] could be adapted to show that accumulating data provides additional stability to the retraining loop and avoids collapse. However, we believe such a study is out of the scope of our work whose aim was to introduce and theoretically develop a new research question of model collapse from the view of preferences. We will, therefore, update our draft to provide additional clarification of this aspect and mention it as an exciting future direction.
>
> I feel like the authors and I miscommunicated here. The point I was trying to raise is: what are realistic assumptions to make about how model-data feedback loops should be modeled? I wasn't so much interested in that other paper as much as I was interested in whether this paper faithfully captures the settings we care about ("I do not think that Equation (3) faithfully models reality. Specifically, it assumes that the -th model is fit to a mixture of (1) real data and (2) preference-filtered data from the -th model. But realistically, synthetic data should amass over time, as I believe is the case with the datasets that the authors mention in their introduction, e.g., LAION-5B.")
>
> My thinking is that the assumptions of this paper are not especially realistic because synthetic data should increase over time and the total amount of data should increase over time too.
>
> **TLDR: I think your assumptions are not realistic and I think this harms the significance & relevance of your paper.**
>
> If I missed the response to this point by the authors, I apologize and I would appreciate being pointed in the correct direction. Thank you!

---

> ### Author Response · Authors · 2024-08-12
> **Response to the Reviewer's Comment [Parts 1 and 2]**
>
> We are grateful to the reviewer for their time and engaging with us during this rebuttal. We answer below the reviewer’s additional questions.
> ## On whether the support of $p_{t+1}$ is included in the support of $p_t$
> We acknowledge the reviewer's comments regarding the fact that the support of $p_{t+1}$ is included in the support of $p_t$. We understand the two points provided by the reviewer as
> >1) Depending on the choice of realizable distributions $\mathcal{P}$, the support might be the entire space of outcomes e.g., if $\mathcal{P}$ is the set of Gaussian distributions.
>
> Note that having that the support is the entire space of outcome is not a problem at all for our theory. Our only requirement is that **the reward is bounded over the support of $p_0$** (at initialization) by $r_*$ (Assumption 2.1) which implies that the reward is bounded for all timesteps (since the support of $p_t$ is included in the support of $p_0$).
> There exist many functions with unbounded support that are bounded (e.g. the sigmoid function).
> We believe that it is reasonable to assume that the intrinsic human reward is bounded.
>
> >2) In general, probabilistic models are often capable of placing mass/density on sets that were not in their training data (because they generalize)
>
> Our setting prevents this scenario from happening: we are working in a setting were we neglect the errors due to the finiteness of **model’s capacity** and **training data**. In other words, we work in a setting where we have:
>
> i.  **infinite capacity regime**:  *the set $\mathcal{P}$ of achievable distributions is the entire set of probability distributions* (lines 112-113 in lemma 2.1). In that case, equation 6 holds and hence the support of $p_{t+1}$ is included in the support of $p_t$.
>
> ii. **infinite training data regime**: This comes from the fact that we are assuming that the next distribution minimizes the population likelihood (and not the empirical one) in equations 3 and 4 ($p_{t+1}$ is defined using an expectation on the distribution $p_t$ and $p_{data}$).
>
> Finally, i), ii) together prevent the points 1) and 2) from happening in our setting.
> Note that we mentioned i) explicitly in the statement of Lemma 2.1 “If $\mathcal P$ is the set of probability distributions on $\mathbb{R}^d$” and on Line 174.
> The point ii) was indicated by the fact that we were using expectations and not finite sum in our paper.
>
> We acknowledge that we should have made i) and ii) appear more clearly in our paper to avoid any confusion. **We will clarify and highlight our setting in the updated manuscript**. Current generative models are getting larger and larger, and are trained on more and more data with more than billions of parameters and datapoints. That is why, we believe it is reasonable to neglect the errors due to the finiteness of **model’s capacity** and **training data** in our theory and that our results satisfyingly capture the reward maximization phenomenon induced by human curation.
>
> ----------
> edit: we updated this comment to enhance clarity of the response

---

> > ### Author Response · Authors · 2024-08-12
> > **Response to the Reviewer's Comment [Part 3]**
> >
> > We are grateful to the reviewer for this interesting discussion on whether our assumptions realistically reflect the practical setting when today’s large models are retrained on web-scaled dataset.
> >
> > We agree together with the reviewer that the accumulation of data is arguably a realistic feature of web-scaled datasets such as LAION-5B. Especially, it is expected that new next-generation web-crawling datasets will incorporate synthetic images from previous years generative models together with the current state-of-the-art generation of generative models. We did not incorporate such feature in our setting as it would complicate the statements and the notations. The main focus of our work was on a new type of collapse centered on human preferences which is different from the previous literature. However, we believe such an extension is relatively easy and we present now how to address it:
> >
> > Let $\\{\lambda_k^t\\}$ for $0 \leq k \leq t$ be a family of positive numbers such that forall k, $\lambda_k^t$ is decreasing in t and normalized such that $\forall t\geq 0, \sum_{k=0}^t\lambda_k^t=1$. Consider that the data accumulates with proportions given by the family $\\{\lambda_k^t\\}$ for $0\leq k \leq t$. In that case, equation 6 which states $p_{t+1}(x) = p_t(x)\cdot H^K_{p_t}(x)$ becomes $p_{t+1}(x) = \sum_{k=0}^t \lambda_k^t p_k(x)\cdot H^K_{p_k}(x)$.
> >
> > We believe that similarly to our theory, it is straightforward that $E_{p_t}[e^{r(x)}]$ is increasing. Further assume that $\forall k, \lambda_k^t\overset{t\rightarrow \infty}{\rightarrow} 0$. In that case, for all $k$ the contribution of each $p_k$ in the retraining at iteration $t$ of $p_t$ decreases to $0$. We additionally believe that **in that case, the expected reward will converge to $r_*$ the maximal reward at initialization and that $p_t$ will collapse to maximal reward regions**.
> > We think that this case is realistic, as *the proportion of data on the web generated by any model is doomed to vanish as more models are trained and deployed*. In that setting, accounting for the accumulation of data would therefore not prevent the collapse of the self-consuming model to maximal reward regions. This constitutes an interesting extension and we will add to the updated manuscript precise statements, along with proofs if the reviewer believes it strengthens our work.
> >
> > Finally, we mention that the work [1] is concurrent with ours since it was first posted on Arxiv on 1st April 2024. This was an additional reason why we did not incorporate their setting in our study, as we were aware of such extension only late in this project.

---

### Official Review · Reviewer_cUrK · 2024-07-13

**Soundness:** 3
**Presentation:** 4
**Contribution:** 3
**Rating:** 7
**Confidence:** 4

**Summary:**

This paper explores the scenario where generative models are iteratively trained on self-generated data curated by human users with some implicit reward. The key idea is that each iteration of training on the self-generated data reweights the previous distribution based on the implicit reward, which converges to reward maximization as the iteration approaches infinity. The paper also studies the scenario where the curated self-generative data is mixed with natural data and analyzes its implication on the stability of iterative training. Experiments on synthetic data and CIFAR 10 validate the insights from the theoretical analysis.

**Strengths:**

1. This paper studies an interesting problem of generative models being iteratively trained on self-generated data curated by human users with some implicit reward. This is arguably an accurate description of what happens when new generative models are trained nowadays.
2. The theoretical framework analyzed the convergence and stability of iterative training with and without reference data.
3. Experiments on synthetic data and CIFAR 10 are interesting, especially the one on CIFAR with replay, which shows how bias amplification can be mitigated with natural data.

**Weaknesses:**

1. Human preference can be heterogeneous, so in some cases, Eq. (2) does not hold. It would be interesting to see if the theoretical analysis can be extended to a mixture of rewards.
2. This paper didn't provide experiments on realistic datasets and large models such as LAION and Stable Diffusion (SD). Thus, it's hard to map the theoretical insights to how we should train the next generation of SD. It would be really interesting if the experiments on CIFAR were reproduced on SD to see what realistic biases are picked up, how much replay is needed to mitigate that, etc.
3. Conceptually, retraining with data curation is related to iterative finetuining in language models (e.g., rejection-sampling-based SFT). Would like to see more discussion on that.

**Questions:**

See Weaknesses.

**Limitations:**

I think the authors adequately addressed the limitations.

---

> ### Author Rebuttal · Authors · 2024-08-06
>
> We thank the reviewer for finding the problem we tackle “interesting” and to be “an accurate description of what happens when new generative models are trained nowadays”. We also appreciate that they find our experiments “interesting”. We now address below the key points raised in the review:
>
> ## Extension to mixture of reward
> Extending our results to hold for a mixture of rewards is a very interesting point that cannot be straightforwardly tackled in our framework. We believe it is an exciting avenue for future work and refer the reviewer to our global response for a more detailed discussion. The crucial difference with our framework is that while we showed that the learned distribution concentrates around the maximal level set of the reward, in the presence of multiple rewards it is not clear to which level set the learned distribution will concentrate.
>
> ## Larger scale experiment
> We acknowledge the reviewer's remark that experiments on larger scales dataset would be interesting to shed light on practical implications of such a collapse in terms of the expected reward. However due to the computational cost of such an experiment, we couldn't perform during the rebuttal period. We would however like to point out that existing studies already show how the style of samples of a generated model evolves when fine-tuned on synthetic and real data with preference optimization. For example, in [1] the authors retrain a generative model by iteratively fine-tuning using Direct Preference Optimization on a mixture of synthetic and real data and where the reward model favors real data. It is demonstrated in Figure 5 [1] that the style of large scale vision models can be shifted by retraining on synthetic data using a preference model. In particular, the model’s interpretation of “a very cute boy” drastically changes, illustrating how the reward influences the realistic biases that are picked up.
>
> ## Relationship with rejection sampling fine-tuning
> We thank the reviewer for encouraging us to strengthen the discussion between iterative finetuning of language models and retraining with data curation. While there is already a large literature on RLHF to iteratively finetune LLMs the reviewer highlights rejection-sampling as one way to optionally see finetuning as a sampling problem amenable to probabilistic inference. Indeed, recent works [2,3] frame iterative finetuning as drawing samples—using rejection sampling, Twisted Sequentional Monte Carlo etc…—from the unnormalized posterior distribution $p(x) \propto e^{r(x)} p_0(x)$,where $p_0(x)$ is an initial generative model trained on real data. From this perspective, our framework studies the case where we curate data by using human reward and obtain $x \sim p(x)$ which are samples from the posterior, without access to the density. This allows us to then finetune $p_0(x)$ to approximate the posterior $p(x)$, which in our notation is a step of iteratively finetuning on curated data. We will include a small discussion on this connection in our updated paper.
>
> We thank the reviewer again for their valuable feedback and great questions. We hope our answer here and in the global response allows the reviewer to consider potentially upgrading their score if they see fit.  We are also more than happy to answer any further questions that arise.
>
> [1] H. Yuan, Z. Chen, K. Ji, and Q. Gu. Self-play fine-tuning of diffusion models for text-to-image generation, 2024.
>
> [2] Zhao, Stephen, et al. "Probabilistic inference in language models via twisted sequential monte carlo.", 2024.
>
> [3] Kong, Lingkai, et al. "Diffusion models as constrained samplers for optimization with unknown constraints.", 2024.

---

### Official Review · Reviewer_WZgs · 2024-07-13

**Soundness:** 3
**Presentation:** 3
**Contribution:** 2
**Rating:** 6
**Confidence:** 4

**Summary:**

Self-consuming generative models are known to have collapse or stability problems, and the curation of synthetic data is often ignored. This paper theoretically studies the impact of data curation and proves that it optimizes the expected reward.

**Strengths:**

1. The paper is well-written, and works on synthetic data are extremely important to the field.
2. The connection between retraining with a mixture of curated data and original data and RLHF (Reinforcement Learning with Human Feedback) with KL regularization is novel.

**Weaknesses:**

1. The theoretical results are applied in a simplified setting, focusing on distribution and reward on $x$ only, instead of considering a joint distribution of $x$ given $y$ mimicking text-to-image generation. The theoretical results apply only to learning the distribution directly, without considering finite samples and optimization, and there is only one reward function. In recent works combining RLHF and diffusion models [1], multiple rewards are considered. How would the results hold with a (random) weighted sum of multiple rewards?
2. On the connection with model collapse: The authors motivate the results from previous literature on synthetic data leading to model collapse but do not discuss how and whether curation with a reward model will avoid model collapse. The reviewer thinks that improvements can occur when inconsistencies between text and image and implausible images are discarded through curation. However, other problems persist in model collapse, such as the inherent lack of diversity in synthetic data [2], and existing results show that human interaction with GPTs still produces data lacking diversity [3]. In this case, curation does not address the loss of diversity in synthetic data. The authors should discuss this clearly, especially since the paper is motivated by model collapse and spends considerable time discussing it.
3. Could the authors explain the contribution of Section 2.2.1? Comparing Sections 2.2.1 and 2.2.2, there seems to be no direct comparison to be made.

### Reference

[1] Liang, Youwei, et al. "Rich human feedback for text-to-image generation." *Proceedings of the IEEE/CVF Conference on Computer Vision and Pattern Recognition*. 2024.

[2] Guo, Yanzhu, et al. "The Curious Decline of Linguistic Diversity: Training Language Models on Synthetic Text." *Findings of the Association for Computational Linguistics: NAACL 2024*. 2024.

[3] Padmakumar, Vishakh, and He He. "Does Writing with Language Models Reduce Content Diversity?." *The Twelfth International Conference on Learning Representations*.

**Questions:**

See weakness.

One related work: [4] propose to use a correction function on the synthetic data to mitigate the model collapse.

[4] Gillman, Nate, et al. "Self-Correcting Self-Consuming Loops for Generative Model Training." Forty-first International Conference on Machine Learning.

**Limitations:**

The limitations and societal impacts are well discussed.

---

> ### Author Rebuttal · Authors · 2024-08-06
>
> We thank the reviewer for their time and review of our paper. We are glad that the reviewer finds our paper "well-written" and our connection with RLHF to be "novel" in the "extremely important" area of work on synthetic data. We address below the key points raised by the reviewer:
>
> ## Extension of the theory
>
> ### Theory for conditional distribution $p(x|y)$
> As remarked by the reviewer, our theoretical results focus only on an unconditional distribution $p(x)$. However we believe that our theory can be easily applied to a conditional distribution $p(x|y)$. Indeed for any sampling of $x_1, \dots, x_K$ given a prompt $y$, and a subsequent curation by humans of these samples, we can apply our theoretical results conditioned on $y$ (especially because the reward $r(x,y)$ also depends on the prompt $y$). In particular, our results state that the expected reward conditioned on $y$, i.e. $E_{p_t(x|y)}[r(x,y)]$ will be maximized and converge to a variable $r^*(y)$ where $r^*(y)$ is given as in our Assumption 2.1 (but now conditionally on $y$). Note that this is valid for any $y$, hence our framework extends to conditional generation.
>
> ### Finite samples and optimization:
> We agree with the reviewer that we work under the assumption of perfect learning. We believe that accounting for finite sample optimization is a crucial extension, but that it would necessitate a more involved framework beyond the scope of our paper, especially in the case of deep generative models optimization. However, we note that a starting point to such analysis could be similar to the assumption 3 made by [5] to include finite sample optimization to their framework.
>
> ### Mixture of reward
> We thank the reviewer for referring us to an interesting related work [1] which investigates the use of rich human feedback to improve text-to-image models. This enhanced human feedback is no longer a single scalar score but consists of multiple evaluation such as delimiting part of an image with implausibility content, or labels on prompts words that are misrepresented in the image. While we proved convergence of $p_t$ to the maximal level set in the presence of a single reward, this raises an interesting question on what dynamics would arise when the maximal level sets of each reward from the mixture are disjoints. Please refer to our global response for more details on mixtures of rewards.
>
> ## Can curation prevent collapse?
> The reviewer highlights an important point related to the difference between collapse of the distribution (i.e. the model only generates a single sample $x$ given $y$), and collapse of the expected reward to its maximum level set.
> - Retraining with curated samples can avoid the former but not later as mentioned L71 of our paper:
> > Retraining with curated samples both maximizes an underlying reward whose variance collapses and converges to maximum reward regions.
> - We also refer the reviewer to the new experiments of the global response, especially Figs.1 and 2 that illustrate how reward’s variance may vanish but not the overall distribution’s variance.
> - However, such a collapse of the expected reward will induce **a decrease of diversity** as the samples with high but non-maximal reward will not be generated. This is additionally shown in our Figs. 7 where we show on CIFAR that, even if the average reward increases, the **FID consistently increases**.
>
> Finally, we mention that we believe that **both the reward and the quality are related since it is expected that low-quality images would have low reward* following human standards and understanding to what extent is an interesting avenue for future works. We will make sure to clarify this point in the updated draft and discuss the two important works [2, 3] mentioned by the reviewer as follows:
>
> In [2], the authors empirically investigate the linguistic diversity of generated text in the self-consuming loop. They consistently observe a decrease in diversity across the retraining iterations. However, unlike our work, they don’t study the impact of curation. In [3], the authors compare the diversity of essays written by humans, assisted either by GPT3, a feedback-tuned LLM (InstructGPT) or without LLM assistance. They found that when using InstructGPT, the overall diversity decreased. As pointed out by the reviewer, this shows that feedback-tuned models have a diversity decrease with respect to the original distribution and that human supervision is not sufficient to compensate for it. It is consistent with our theoretical insights: we show how retraining with curation incurs a collapse of the reward’s variance similar to feedback fine-tuning.
>
> ## Sec. 2.2.1
> As rightfully pointed out by the reviewer, Sec. 2.2.1 revisits the framework proposed by [5] which differs from the rest of the section. We chose to incorporate this section as a preliminary to the rest of the section for the following reasons:
> - It introduces the reader to how reusing real samples in the retraining loop can provide stability.
> - It formulates the stability results of [5] using a Kullback-Leibler divergence term, which is new and is the same formulation as in Thm 2.4.
> - It yields **a significant improvement on prior work** for the upper bound on the parameter $\lambda$ which [5] left as future work.
>
> [5] Bertrand, Q., et. al. On the Stability of Iterative Retraining of Generative Models on their own Data. In ICLR 2024
>
> We will make the setting distinction clearer in the updated draft to avoid confusion.
>
> ## Related work
> We thank the reviewer for this reference to [4]. We note that this work is presently cited on line 47 but are happy to provide more discussion.
>
> We are grateful to the reviewer for their great questions and hope that our answers in conjunction with the global response clarified them. We politely encourage the reviewer to ask any further questions they may have and, if they are presently satisfied with our responses, to consider a fresher evaluation of our paper.

---

> > ### Comment · Reviewer_WZgs · 2024-08-11
> >
> > I appreciated the author's discussion on the multi-reward setting and on the difference between the collapse of distribution and the collapse of reward variance. I have increased my score.

---

### Author Rebuttal · Authors · 2024-08-06

We would like to thank all the reviewers for their detailed feedback which has allowed us to strengthen the updated manuscript. In particular, we are heartened that all reviewers (WZgs, cUrK, ej3h, cgiY, orKH) found our research question to tackle an interesting, highly-sensible, and timely problem. We are also glad that reviewers (WZgs, ej3h) appreciated the writing of the paper. We below address some key points that were raised in multiple reviews:

**Extension of our framework to using a mixture of rewards (reviewers WZgs, cUrK):**

We thank the reviewers for raising a very interesting point regarding the use of mixed rewards. Frameworks going beyond a single reward model are especially relevant in practical scenarios in LLM alignment. An interesting reference on this topic is the recent work [1] which addresses such extension by learning a preference model of samples given a prompt $P(x ≻ x’|y)$ (as a function of the two variables $x, x’$)--- instead of the Bradley-Terry reward model $r(x)$ (less general when preferences are non-transitive) which they refer to as Nash Learning from Human Feedback.

We now outline how to derive an extension to a mixture of reward to our setting:
First, we can introduce a new latent variable $u$ that describes the randomness in the reward used, which leads to the following expression of the curated distribution after one step of curation:

$$
p_{t+1}(x) = p_t(x) \cdot H^K_{p_t}(x) \quad  \text{with} \quad  H^K_{p_t}(x):= E_{x_1,\ldots,x_{K-1} \sim p_t, u} \left[\frac{K \cdot e^{r(x; u)}}{e^{r(x; u)} + \sum_{i=1}^{K-1}e^{r(x_i; u)}} \right]
$$

In our setting, we were able to prove that the expected reward increases and the distribution converges to the maximum level set of a unique reward (Lem 2.2, Thm 2.1). However, in the presence of multiple rewards, it is not straightforward that the rewards have the same maximal level sets. Therefore this may yield interesting dynamics and the convergence of $p_t$ may differ. We believe such an extension of our results is outside of the scope of this work and think that it is a fascinating avenue for future work. For example, it may be interesting to study if a reward component in the mixture dominates, thereby dictating the convergence, e.g. if it gets large differences between two samples. In that case, the distribution may converge to only one maximal level set introducing a new model collapse behavior as the mixture of rewards would be dictated by a single reward. We added this discussion on the multiple rewards setting and [1] our revised manuscript.

**Extended related work and contextualization of our results in the literature** We thank the reviewers for inviting us to develop our discussion of related works. We will include in the updated paper an in-depth discussion on rejection sampling finetuning (cUrK, ej3H, cGiY), RLHF (ej3H) and comparison of our setting with respect to the previous model collapse literature (orKH).

**Clarification and additional experiments** We thank reviewers orKH and ej3h for suggesting improvements to our experiments which we believe will help clarify them and illustrate the theory better. Especially, our new experiment in Figures 1 and 2 of the figures pdf shows that the reward variance may vanish independently of the overall distribution variance. This underlines the crucial difference between model collapse from the view point of previous works and collapse of the reward variance, that we introduce and develop in our work.

## References

[1] Munos, Rémi, et al. "Nash learning from human feedback." 2023.

---

### Decision · Program_Chairs · 2024-09-25

**Decision:**

Accept (spotlight)

**Comment:**

This paper performs a theoretical analysis of the setting in which a generative model is trained on a curated dataset of its own samples.  It shows that the implicit reward that guides such curation can successfully be learned by the generative model, thus motivating the soundness of the practice.  The reviewers were unanimously in favor of acceptance, with the paper's strengths being that it performs a novel, rigorous theoretical analysis of a topic of great modern importance.  Moreover, the results are experimentally supported in interesting experiments (Reviewer cUrK).  The reviewers primary concerns were if the analysis extends to a mixture of reward functions, relationship to prior work, and on some of the underlying conditions / assumptions of the results.  I believe the authors have either successfully addressed  these concerns (e.g. the underlying conditions) or rebutted their significance (e.g. mix of rewards is a nice extension but not essential, some of the related works are concurrent or specialized to specific generative model classes).